# Experimental demonstration of highly reliable dynamic memristor for artificial neuron and neuromorphic computing

See-On Park [1,2], Hakcheon Jeong [1,2], Jongyong Park[1,2], Jongmin Bae[1] & Shinhyun Choi [1✉]

Neuromorphic computing, a computing paradigm inspired by the human brain, enables energy-efficient and fast artificial neural networks. To process information, neuromorphic computing directly mimics the operation of biological neurons in a human brain. To effectively imitate biological neurons with electrical devices, memristor-based artificial neurons attract attention because of their simple structure, energy efficiency, and excellent scalability. However, memristor's non-reliability issues have been one of the main obstacles for the development of memristor-based artificial neurons and neuromorphic computings. Here, we show a memristor 1R cross-bar array without transistor devices for individual memristor access with low variation, 100% yield, large dynamic range, and fast speed for artificial neuron and neuromorphic computing. Based on the developed memristor, we experimentally demonstrate a memristor-based neuron with leaky-integrate and fire property with excellent reliability. Furthermore, we develop a neuro-memristive computing system based on the short-term memory effect of the developed memristor for efficient processing of sequential data. Our neuro-memristive computing system successfully trains and generates bio-medical sequential data (antimicrobial peptides) while using a small number of training parameters. Our results open up the possibility of memristor-based artificial neurons and neuromorphic computing systems, which are essential for energy-efficient edge computing devices.

[1] The School of Electrical Engineering, Korea Advanced Institute of Science and Technology (KAIST), Daejeon 34141, Republic of Korea. [2] These authors contributed equally: See-On Park, Hakcheon Jeong, Jongyong Park. ✉email: shinhyun@kaist.ac.kr

Artificial neural networks (ANNs) have shown their effectiveness in various fields such as autonomous cars, finding new drugs, designing circuits, or predicting protein structures[1–4]. Powered by today's elaborate complementary metal–oxide-semiconductor (CMOS) based computing processors, ANNs can accomplish highly complex tasks. However, ANNs running in a conventional von Neumann architecture computer (e.g., graphics processing units (GPUs)) cannot reach the efficiency of a biological neural network due to the bottleneck of big data transfer[5]. To substitute the conventional ANNs, neuromorphic computing hardwares, which imitate the operation of the brain and the biological neural network, have been extensively studied. The conventional CMOS-based neuromorphic hardware uses complex CMOS circuitry to imitate the neuronal activity[6]. However, CMOS-based complex circuitry limits scalability and increases energy consumption. Thus, it is difficult to reach the efficiency of a biological neural network using CMOS-based neuromorphic hardware[7].

Memristors, instead of the CMOS-based circuit, have been widely studied as a new candidate for neuromorphic hardwares. Memristors are emerging memory devices that store information in the form of the resistance by changing the internal distribution of oxygen anions or metal cations[8,9]. They are a promising candidate for an artificial neuron devices for neuromorphic hardwares because of their great scalability, high energy efficiency, fast speed, small footprint, and simple fabrication process. Previous studies have developed several memristive artificial neurons and proved that memristors are proper devices for efficient artificial neurons[5,7,10,11]. Short-term (volatile) memristors have been widely used to represent a leaky-integrate and firing (LIF) property of a biological neuron, instead of being used for a synapse. Short-term memristor-based artificial neurons have been used for various neuromorphic computing systems such as neuro-memristive computings and spiking neural networks (SNNs)[10,12–14].

Despite these advantages, the current memristor-based artificial neuron has several limitations. Various studies have proven the feasibility of memristor-based neurons by unambiguously showing neuronal operations. Among them, diffusive memristors ($SiO_xN_y$:Ag) or Mott memristors ($NbO_2$ or $VO_2$) have been extensively studied as artificial neuron devices[7,14,15]. The diffusive memristor type neurons are fast and have high on/off ratio, and the Mott memristor type neurons have acceptable uniformity and fast speed. However, the diffusive memristor type neurons have uniformity issues, and the Mott memristor type neurons require a large operation current (~mA) and have small on/off ratio. In general, memristors usually suffer from large variation problems and unreliable switching[8,9,16]. These reliability problems degrade the memristor-based neuron's characteristics and make it difficult to build a large-scale memristor-based neuromorphic computing hardware. Even though reliable memristors may exist, it is difficult to operate a memristor-based cross-bar array without other components, such as selectors and transistors, because undesired current paths, which are called sneak paths, hinder the reliable read and set/reset operations in a memristor cross-bar array[8]. In addition, the low on/off ratio increases the complexity of the peripheral circuit that is needed to distinguish the small conductance change of the memristor, and high current requirement (~mA) increases energy consumption. To overcome these bottlenecks, development of a new memristor that satisfies all of the requirements is warranted.

Here, we propose a transistor-free 1R structure memristor that consists of a metal oxide with gradual oxygen concentration that are fabricated in low-temperature environments for a memristor-based artificial neuron and a neuro-memristive computing system construction. We demonstrate that this memristor performs with high yield in array form (~100%), obtains self-rectifying behavior, has high temporal/spatial uniformity (1.39% and 3.87%, respectively), high endurance without degradation (>10^6), high speed (10 μs), high on/off ratio (>2000), and uniform decaying time constant in the array. Based on these ideal properties, we show that the developed memristor-based artificial neuron possesses the leaky-integrate and fire (LIF) characteristic, which is the key characteristic of a biological neuron. The developed neuron has high spatio-temporal uniformity, which is one of the essential features for building a reliable memristor-based neuromorphic hardware. In addition to the demonstration of the artificial neuron, we build a neuro-memristive computing system by using the memristors as leaky-integrate neurons. The developed neuro-memristive computing system can deal with sequential data, which can provide further complications compared to temporal data processing. With the developed neuro-memristive computing system, antimicrobial peptides (AMPs), the anti-bacterial elements in the innate immune system, are utilized. The neuro-memristive computing system trained by AMP data successfully learns the complex amino-acid grammar of AMPs and generates new AMPs.

## Results

**Characteristics of gradual $TiO_x$-based memristor.** The memristive device with gradual $TiO_x$ layer is fabricated via an anodizing process (see "Methods" section), as shown in Fig. 1a. The device shows only a 1.39% temporal variation (σ/μ) during 125 consecutive DC cycles without current compliance (see Fig. 1b). The I–V curve also shows that the device has a high rectifying ratio of $10^4$ and high on/off ratio larger than $10^3$. This high rectifying ratio (current at $V_{read}$/current at $-V_{read}$) prevents the selector-less cross-bar array from the sneak path current problems (see Supplementary Fig. 1), and the high on/off ratio (>$10^3$) reduces the peripheral circuit burden to discriminate the conductance change, which are the major bottlenecks in conventional memristor cross-bar arrays. Because of the forming-free nature of the device, where the forming process usually damages the devices' performance[17], the device in the memristor cross-bar array achieves high endurance of more than $10^6$ cycles without any degradation of the electrical performance (see Fig. 1c and Supplementary Fig. 2). These forming-free and compliance current-free properties of gradual $TiO_x$ memristors make the device more suitable for neuromorphic hardware in edge devices with simple circuit designs[18]. The gradual $TiO_x$ memristor shows exceptionally uniform resistive switching with only a 3.87% spatial variation (σ/μ) from 30 randomly selected devices, as shown in Fig. 1d. The memristor device also exhibits a low temporal variation of about 1.67% (σ/μ) during 50 cycles of 200 consecutive set pulses (4.5 V, 10 μs) and read pulses (1.5 V, 100 μs) (see Fig. 1e). These results show the high uniformity of the developed device, which has been a significant bottleneck for various memristor devices. The uniform 40 ms time constant (τ) during the self-decaying process is also achieved with a low temporal variation of 3.86% (σ/μ), which is one of the key points for a reliable artificial neuron and neuro-memristive computing system.

Finally, all 400 devices from a $20 \times 20$ cross-bar array (see Supplementary Fig. 3) are measured to show spatial uniformity in terms of time constant and conductance change as a function of the number of applied pulses. The uniformity of the decaying time constants from a cross-bar array is shown in Fig. 1f. The time constants measured from the 400 memristors are ~40 ms, which is similar to the time constant calculated from the stand-alone device shown in Fig. 1e. This means that all the devices in an array operate uniformly, and due to the self-rectifying behavior, the read process is not disturbed by the sneak path

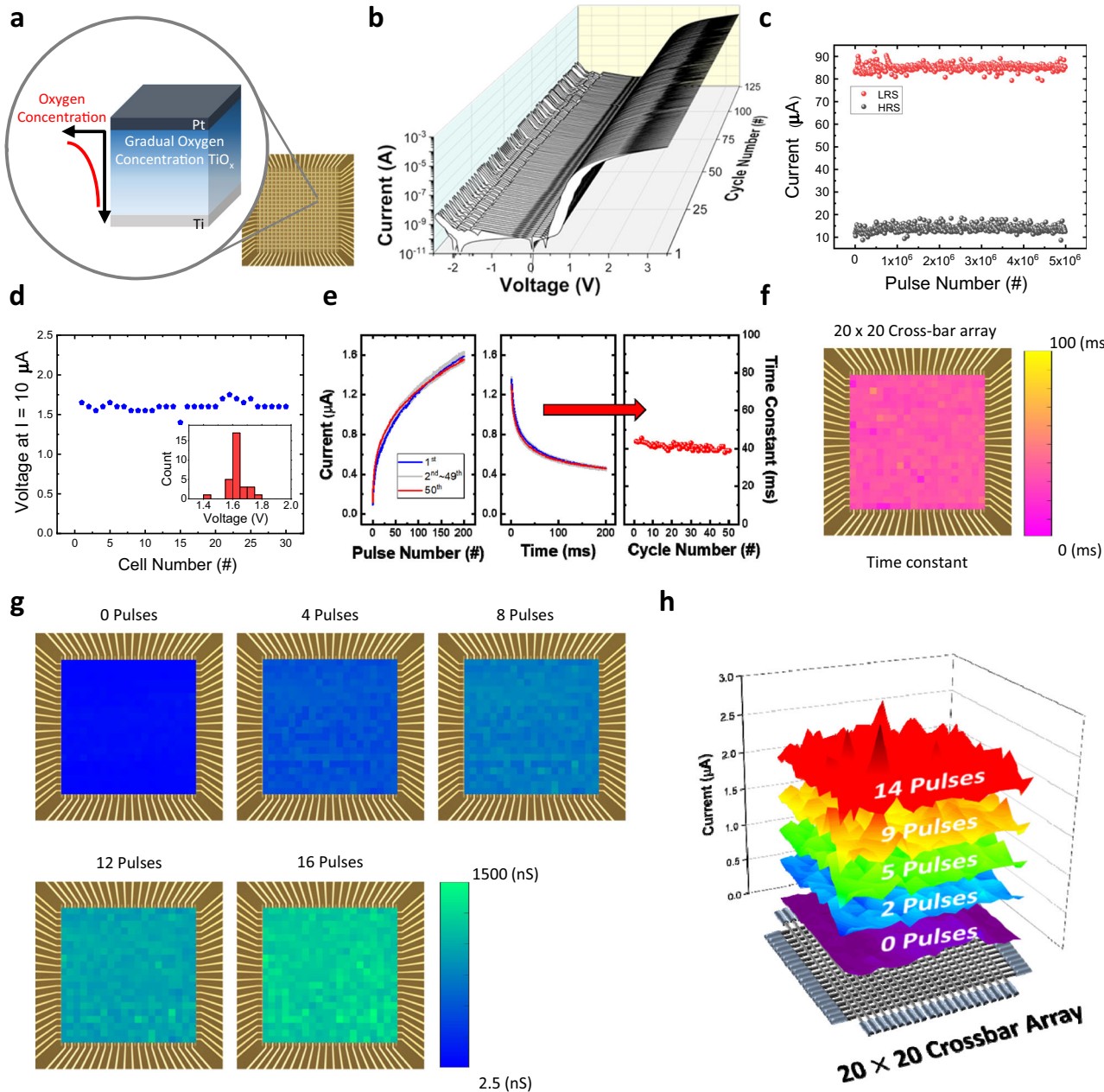

**Fig. 1 The structure of gradual TiO$_x$ memristor and its electrical characteristics. a** Schematic of Pt/gradual TiO$_x$/Ti memristor structure showing gradual oxygen concentration in the TiO$_x$ layer. **b** Consecutive 125 I–V sweep curves of the device showing very small temporal variation (1.39%) during the cycles. **c** Endurance test showing 5 × 10$^6$ consecutive set/reset pulse without degradation. **d** Spatial variation (3.87%) calculated through DC voltage sweep in the 30 stand-alone devices. **e** Fifty consecutive cycles of the pulsed response of a single stand-alone device. A single cycle is composed of 200 set pulse responses (4.5 V, 10 μs) and following self-decaying during 200 ms with a 40 ms decaying time constant. **f** A 2D map for the decay time constant of 400 devices in a 20 × 20 cross-bar array. **g** Analog conductance change in a 20 × 20 cross-bar array with 0, 4, 8, 12, and 16 set pulses (4.5 V, 100 μs). **h** A 3D representation of the conductance change along with the set pulses.

problem. Even though the device cannot serve as a long-term memory due to the short-term characteristics, the short-term memory characteristic with excellent uniformity makes the developed memristor suitable for artificial neuron and neuro-memristive computing. As shown in Fig. 1g, the device changes its conductance from 55 nS to 1200 nS (on average) with small spatial variations by applying 16 write pulses. To show the potentiation of the memristors with only 16 pulses, a longer set pulse (4.5 V, 100 μs) is used, and the memristor conductance is measured by read pulses (2 V, 100 μs). The conductance map as a function of the device's location in the optical image of the array shows uniform distribution conductance change, as

shown in Fig. 1h. The conductance changes during the 16 pulses are represented in Supplementary Fig. 4.

**Insulator thickness modulation in gradual TiO$_x$.** A cross-sectional transmission electron microscopy (TEM) image of a gradual TiO$_x$ memristor is shown in Fig. 2a. A thin gradual TiO$_x$ layer (about 30 nm) is sandwiched by the top electrode (TE) and the bottom electrode (BE), showing a simple structure of the developed memristor. As shown in Fig. 2a, the anodized TiO$_x$ layer does not have any porous features, even though the anodizing technique is widely used to form nanoporous TiO$_2$ by

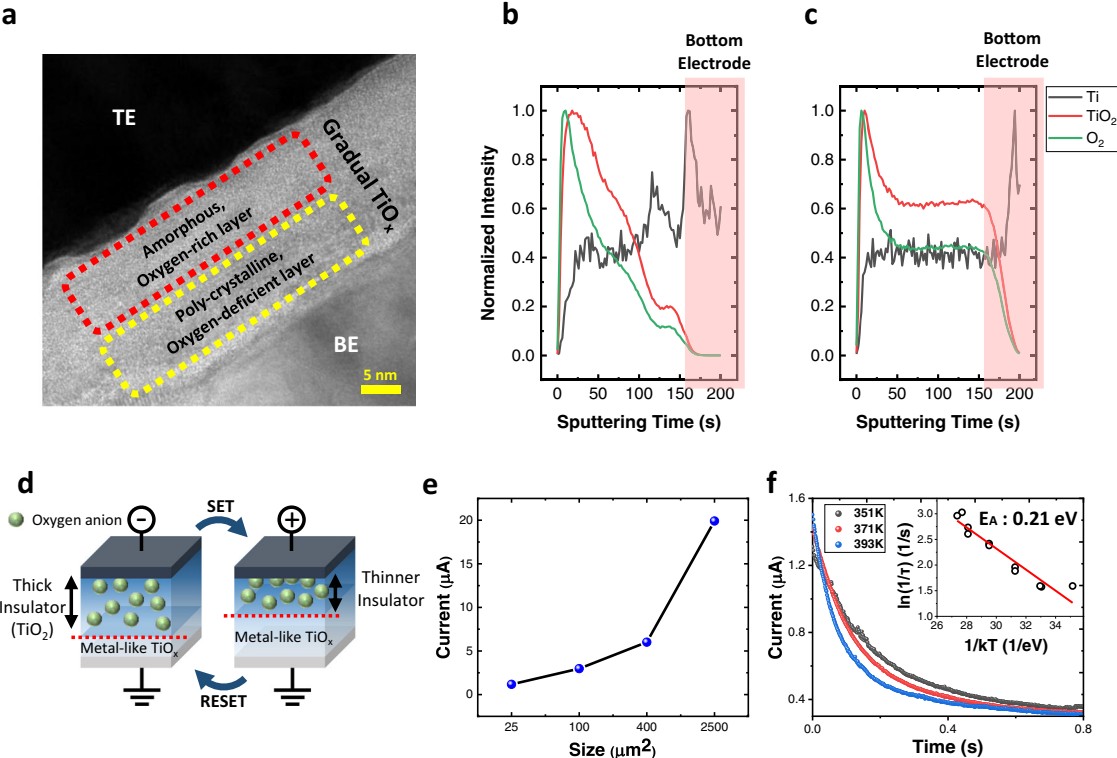

**Fig. 2 Effects of gradual oxygen concentration in TiOx switching layer. a** A cross-sectional TEM image of the gradual TiOx memristor. **b** TOF-SIMS depth profile results of anodized TiOx. **c** TOF-SIMS depth profile results of sputtered TiO2. **d** A schematic representation of the switching mechanism of the gradual TiOx memristor. **e** Size dependency of the gradual TiOx memristor. The memristors with four different sizes ($5 \times 5$, $10 \times 10$, $20 \times 20$, and $50 \times 50$ [$\mu m^2$]) are fabricated on the same wafer. The current is measured at 1 V at a low resistance state (LRS) during the linear voltage sweep from 0 V to 4 V. **f** An activation energy ($E_A$) calculated from the time constant under various temperatures. The time constant decreased when the temperature increased, and the activation energy calculated from the change of the decay time along the temperature implies that the resistive switching and short-term dynamics originate from the oxygen anions in TiOx.

anodizing the Ti film in a fluoride-rich electrolyte. The time of flight secondary ion mass spectrometer (TOF-SIMS) results reveal the difference between the anodized TiOx layer and the radio frequency (RF)-sputtered TiO2 layer. As shown in Fig. 2b, for an anodized TiOx layer, the oxygen concentration and TiO2 concentration gradually decrease from the top (Pt/TiOx interface) to the bottom (TiOx/Ti interface), while the Ti concentration increases. However, these concentration gradients are not observed for the sputtered TiO2 layer (see Fig. 2c). This is because the anodizing process is a top-down oxidation process; therefore, the upper Ti layer undergoes longer oxidation and becomes an oxygen-rich insulating TiO2 layer, while the bottom Ti layer experiences a shorter oxidation time and forms an oxygen-deficient metallic TiOx layer. This oxygen gradient effectively controls the thickness of the insulating region and results in resistive switching with a large on/off ratio of the anodizing TiOx layer, while the sputtered TiO2 layer cannot effectively change the thickness of the insulating region due to uniform oxygen concentration and shows a low on/off current ratio and a non-uniform switching characteristic (see Supplementary Fig. 5).

To be more specific, the oxygen vacancy gradient enables resistive switching with a self-decaying property (see Fig. 2d). When a positive bias is applied to TE while grounding BE, oxygen anions shift from TiOx near BE to TiOx near TE. TiOx near BE loses its oxygen and the metallic TiOx region increases. Therefore, the effective thickness of insulating TiOx decreases, the reduced effective insulating thickness increases the overall device's conductivity, and the device becomes a low resistance state (LRS). However, when a negative bias is applied to TE, oxygen

anions in oxygen-rich TiOx near TE shift to TiOx near BE. The effective insulating thickness expands again, and the device becomes a high resistance state (HRS) (see Supplementary Fig. 6). The LRS to HRS transition (reset) occurs without external stimulus due to the diffusion of oxygen anions, which results in short-term (volatile) memory effects with about 40 ms time constant.

To demonstrate that the device switches its resistance state through the oxygen anion shift in the overall TiOx bulk, the currents from memristors of different sizes are compared, as shown in Fig. 2e. During the linear voltage sweep, the current of the memristor at 1 V (LRS) is read. The results show that the current increases as the size increases. When the memristor operates through a conductive filament, the memristor does not show size dependency, as the locally formed filament conducts almost all of the current regardless of the device's size. However, when the memristor switches its resistance through the overall bulk, the size affects the current. The size dependency in the results verifies that the gradual TiOx memristor operates by modulating the oxygen anion distribution in the overall TiOx bulk. In Supplementary Fig. 7, the device current reduces as the device size decreases due to the non-filamentary nature of the device. Moreover, it does not significantly change other properties such as decaying speed and potentiation ratio, which means that the device is scalable without performance degradation.

Furthermore, by calculating activation energy ($E_A$) from the decaying time constants under several elevated temperatures from 323 K to 423 K, we have proven that the oxygen anion

movement across the whole TiO$_x$ bulk dominates resistive switching (see Fig. 2f). The calculated activation energy ($E_A$) is about 0.21 eV, which is similar to the activation energy of the ionized oxygen vacancy in polycrystalline H$_2$O-rich TiO$_2$[19]. The calculated activation energy of the gradual TiO$_x$ memristor is much smaller than the oxygen diffusion activation energy (1.05 eV) in rutile TiO$_2$[20]. This is because H$_2$O molecules, used during the anodizing process, significantly reduce the activation energy[19]. Moreover, in the anodized TiO$_x$ layer, it is observed that the upper region of the TiO$_x$ has a fully amorphous structure while the bottom region has a polycrystalline structure, because polycrystalline metallic Ti concentration is much higher in the bottom region of the TiO$_x$ layer (see Supplementary Figs. 8, 9). The oxygen activation energy is low in the case of the amorphous phase[21], thus the gradual TiO$_x$ layer has much smaller oxygen anion activation energy compared to the rutile TiO$_2$ layer. This small activation energy allows the oxygen anions to easily move along the bias and induces resistive switching with a large on/off ratio and short-term memory effect. All of these results demonstrate that resistance switching in gradual TiO$_x$ is induced by the movement of oxygen anions across the whole TiO$_x$ bulk, unlike other filamentary-type memristors.

**Reliable LIF neuron with gradual TiO$_x$ memristor**. A biological neuron receives spike signals from presynaptic neurons and integrates the signals into its membrane potential (see Fig. 3a). When a neuron receives spikes in a certain interval, the neuron's membrane potential is elevated. Once the membrane potential exceeds the threshold potential, the neuron sends spikes (an action potential) to its post-synaptic neuron. However, because the neuron has leaky-integrate property, the membrane potential goes back to its original potential (a resting potential) if the presynaptic neuron stops sending spikes and the membrane potential does not reach the threshold. The leaky-integrate and fire (LIF) neuron model elaborately describes these biological neuron's properties[22].

To demonstrate the gradual TiO$_x$-based memristor as an artificial neuron device, the gradual TiO$_x$ memristor-based LIF neuron is constructed with a parallel capacitor and a serial resistor (see Fig. 3b, c). The parallelly connected volatile memristor and capacitor operate as a LIF neuron's soma, where the biological neuron integrates spikes and makes an action potential when the membrane potential reaches the threshold. The resistor (or non-volatile memristor), that serves as a synaptic weight, is serially connected to the gradual TiO$_x$-based volatile memristor and capacitor[10,14]. To examine the neuron operation without non-ideality factors from the synapse device, a fixed resistor is utilized. The resistance is low if the synaptic weight is high, and the resistance is high if the synaptic weight is low. If the parallel capacitor integrates enough charges from presynaptic spikes in a certain time interval, the capacitor voltage becomes high enough to potentiate the volatile memristor. For example, the LIF neuron operation of the gradual TiO$_x$ memristor-based artificial neuron with a 47 kΩ resistor and a 10 nF capacitor is shown in Fig. 3b. After applying two presynaptic voltage pulses, the neuron reaches the threshold voltage and fires.

The artificial neuron's characteristics in terms of input spike frequencies are investigated in Fig. 3d, e, and f. The two different frequency spike trains (3200 Hz and 1600 Hz) are applied to the memristor-based artificial neuron. When the frequency is high, the neuron fires within three spikes. However, the neuron does not fire if the input spike frequency is low at 1600 Hz. This is because the developed neuron has leaky-integration property, and the capacitor voltage does not reach the threshold voltage.

Furthermore, in a biological neuron, the neuron easily fires when it receives spikes from a strongly connected presynaptic neuron, while it requires much more spikes to fire when the presynaptic neuron is weakly connected. This biological phenomenon is emulated in our artificial neuron by modifying the synaptic weight from the serial resistor (see Fig. 3b, g–i). When an artificial neuron is composed of a gradual TiO$_x$ memristor, a serial resistor (47 kΩ), and a parallel capacitor (10 nF), three presynaptic spike pulses are needed to make the neuron fire (see Fig. 3b). If the serial resistance increases to 94 kΩ, that represents a weak connection to the presynaptic neuron, the artificial neuron needs six consecutive spikes to fire (see Fig. 3g). On the other hand, if the serial resistance decreases to 23.5 kΩ representing a strong synaptic connection, the artificial neuron fires within two spikes as shown in Fig. 3h. In a human's biological neural network, the synaptic weight is long-term potentiated when the post-synaptic neuron fires just after the presynaptic neuron fires, because it means the post-synaptic neuron's firing event is strongly related to the presynaptic neuron's firing. This biological phenomenon is called a "Hebbian learning rule", and this is considered as a basic learning mechanism of biological neural networks[23]. The results show that the memristor-based artificial neuron behaves as the biological LIF neuron, and therefore, it can be utilized for reliable neuromorphic processors with biological learning rules.

To build a reliable neuromorphic processor with memristor-based neurons, highly reliable neuron operation is required. In Fig. 3j, k, the gradual TiO$_x$ memristor-based artificial neurons show similar neuron responses, which means the proposed memristor-based artificial neuron has low device-to-device variation. The identical spike trains are applied to the randomly selected 22 artificial neurons, and the outputs of the neurons are shown in Fig. 3j. Every neuron operates in the same manner, without any noticeable differences. Moreover, we also test the cycle-to-cycle uniformity by applying the same pulse train 50 times to a single artificial neuron (see Supplementary Fig. 10). The response of the artificial neuron is not significantly different, demonstrating its low cycle-to-cycle variation. The low variation of the artificial neuron originates from the reliable switching behavior of the gradual TiO$_x$ memristor.

To further investigate the characteristics of the gradual TiO$_x$ memristor-based artificial neuron, the artificial neuron's responses are measured in terms of presynaptic spike amplitude and presynaptic spike width (see Supplementary Figs. 11, 12). As shown in Supplementary Fig. 11, the artificial neuron does not fire with a small amplitude of the presynaptic spikes (2.5 V), but it fires with a large amplitude of the spikes (>3.5 V). In Supplementary Fig. 12, it is demonstrated that the neuron can operate with a short pulse (40 µs). Moreover, the artificial neuron's responses with different combinations of the circuit components are measured; (1) different capacitance values and (2) different capacitance and resistance values while keeping the same R × C value (see Supplementary Figs. 13, 14). The developed artificial neuron has a great tunability to adjust the threshold, by changing the parallelly connected capacitor or the serially connected resistor.

The gradual TiO$_x$ memristor-based neuron demonstrates superior device-to-device and cycle-to-cycle uniformity, tunability, low spike current (10–35 µA), high off-state resistance (~100 MΩ), and an acceptable speed (>40 µs). The comparison table for various memristor-based artificial neurons is shown in Supplementary Table 1.

**Neuro-memristive computing for AMP generation with gradual TiO$_x$ memristor cross-bar array**. Here, based on the short-term memory effect of the gradual TiO$_x$ memristor, the

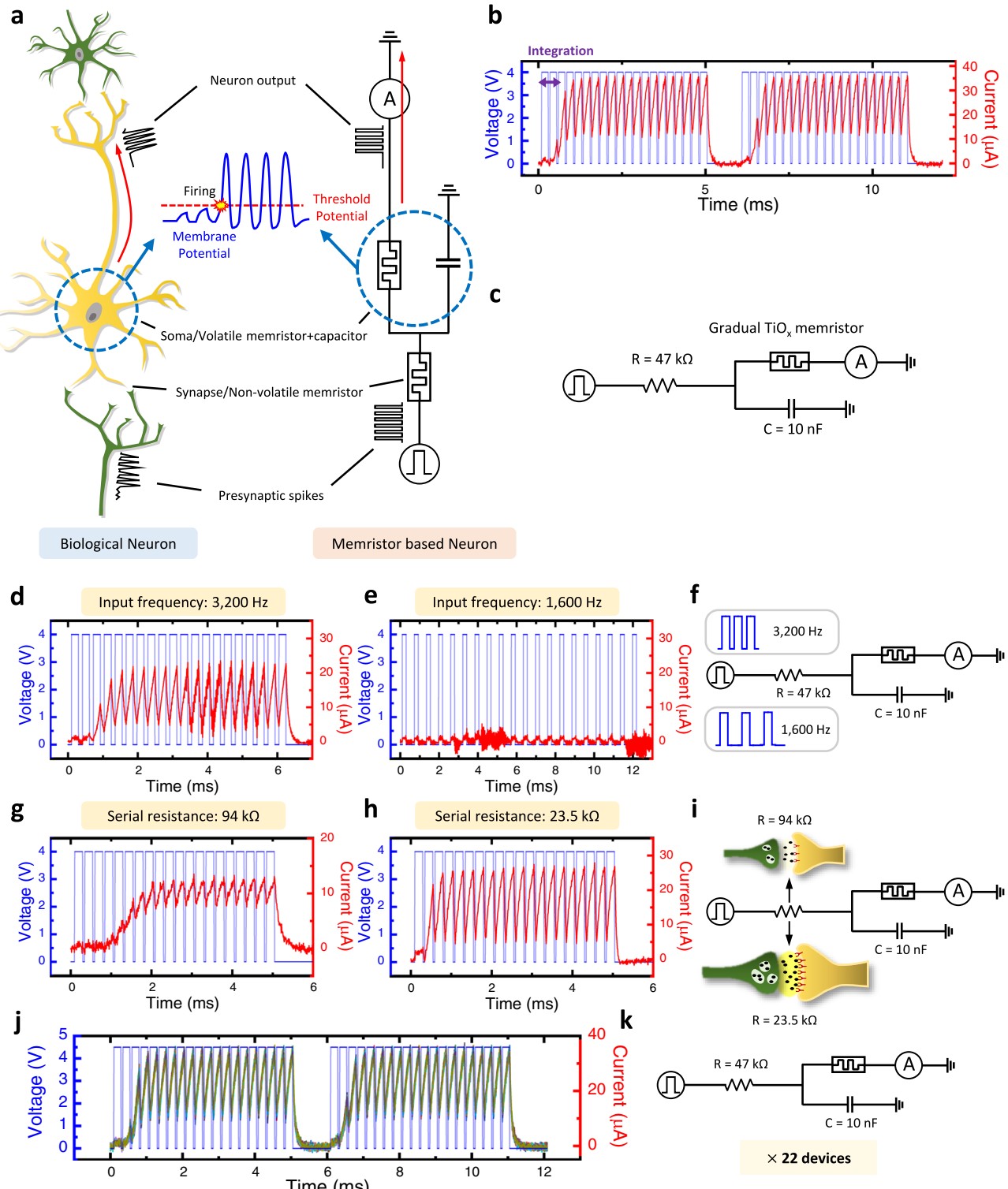

**Fig. 3 Gradual TiO$_x$ memristor-based LIF neuron. a** An illustration of a biological neuron and the analogous memristor-based artificial LIF neuron. **b** The LIF neuron operation of the developed memristor-based artificial neuron. Applied spike train condition: 4 V, 200 μs width, and 50 μs interval. **c** The schematic of the circuitry used for the gradual TiO$_x$ memristor-based artificial neuron. **d–f** The output of the artificial neuron along with different presynaptic spike frequencies (3200 Hz (**d**) and 1600 Hz (**e**)), and the illustration of the experimental setup. During the test, the pulse amplitude and width are fixed (4 V, 200 μs). **g–i** The output of the artificial neuron along with the different synaptic strengths (different resistors). When the synaptic strength is weak (large resistance (**g**)), the artificial neuron fires within six spikes, while it requires fewer spikes when the synaptic strength is strong (small resistance (**h**)). **j**, **k** The outputs of the 22 different artificial neurons demonstrate low variation of the gradual TiO$_x$ memristor-based artificial neuron.

developed memristor is used as a leaky-integrator to build a neuro-memristive computing system which processes sequential data. When the voltage pulse train is applied to the short-term memristor, the conductance state of the device at a specific time is determined simultaneously by a pulse at that time and pulses in the near past around the device's decaying period, similar to a membrane potential of a neuron. This dynamic response of the short-term memristor about the pulse train can be utilized as a sequential/temporal data processor that separates the different sequential or temporal data, which requires complex calculations to process with software algorithms. Based on the sequential/temporal data processing ability of short-term memristors, a simple neural network, which is called a read-out function, can easily learn the input data with a very small number of parameters compared to conventional long short-term memory (LSTM) model[24–26].

Similar approaches to process sequential/temporal data with volatile memristors[12,13,27–29] have been extensively studied, but there have been several limitations. For example, large variation and unreliable switching of the memristor disturb the training of the system, and prevent the neuro-memristive computing system from being used for inference tasks because every memristors will respond differently to the same input and the readout function must be re-trained for every different memristors. Other endemic problems of memristors such as sneak path and low on/off ratio also impede the realization of efficient neuro-memristive computing based on the memristor cross-bar array.

The gradual $TiO_x$ memristor, however, shows several ideal characteristics such as high uniformity, high on/off ratio, and self-rectifying property, and it can be integrated in a cross-bar array structure having high scalability. Therefore, to realize an efficient neuro-memristive computing system without the mentioned problems, the sequential data processing abilities of highly reliable gradual $TiO_x$ memristor are examined in Fig. 4. All experiments are conducted in a $20 \times 20$ gradual $TiO_x$ memristor cross-bar array, where a single cell size is $5 \times 5 \, \mu m^2$. First, for sequential data analysis, the pulse trains for the combination of 4-bit sequence inputs are applied to the randomly selected 24 memristors in a cross-bar array, and the results are shown in Fig. 4a. Every memristor separates a different combination of the input data at the final time step (time step = 4). Each memristor shows an identical and uniform response to the 4-bit input due to the excellent uniformity of the gradual $TiO_x$ memristor. This experiment proves that the gradual $TiO_x$ memristor as a leaky-integrate neuron can separate different binary sequences with high uniformity in an array platform. It is noticeable that the devices show high on/off ratio (~20) with only four pulses in Fig. 4a. The maximum on/off ratio from the developed memristor's large on/off ratio (>2700×), as shown in Supplementary Fig. 15.

We have further investigated how the gradual $TiO_x$ memristors process general sequence data, as shown in Fig. 4b. Two differently ordered alphabet sequence data consisting of eight letters (from "a" to "h") are used with a cross-bar array, as shown in Supplementary Fig. 16. The sequences are converted to a pulse train and the pulse train is fed into the memristors. The memristor assigned to the character is potentiated by the pulse, while the others are rested and naturally decayed. Because of the self-decaying (short-term memory) property, the latest pulses strongly affect the memristors' current states, while the former pulses hardly affect the current state, as the information is already lost due to the decaying effect[12]. Therefore, the information on sequence history can be represented as the conductance level of the assigned memristors. The final conductance states of all memristors are obtained at the final

time step. As shown in Fig. 4b, the outcome of the memristors distinguishes the differently ordered alphabet sequences.

Interestingly, the duty cycle (for write pulse only) controls the decaying speed and changes the response of the memristor for the same input (see Supplementary Figs. 17, 18). As shown in Supplementary Fig. 17, a high pulse width/interval (W/I) ratio results in slow decaying speed (long decaying time constant), while a low pulse W/I ratio results in a fast decaying speed (short decaying time constant). The decaying time constant can be controlled from about 15 ms to 40 ms by varying the W/I ratio, and the different decaying speed enables the memristors to process the same input differently. A comparison between high and low (W/I) ratio cases is represented in Fig. 4d–f. As an example, an alphabet sequence composed of "A", "B", "C", and "D" ([first] D D D C C B D C C D B A D B A C D C B A [last]) is used, as shown in Fig. 4c. In the sequence input, "A" is applied to the memristor at last, and "B", "C", and "D" follow. However, contrary to the entering order, "D" appears in the sequence seven times and shows the highest frequency, while "C" (six times), "B" (four times), and "A" (three times) follow. In the high W/I ratio (W/I ratio = 85/15) case, the frequent inputs dominantly affect the current state of the memristors because of the long decaying time constant (see Fig. 4f). As a result, the most frequently appearing alphabet "D" wins all elements. Furthermore, in the short-term (W/I ratio = 15/85) case, the recent inputs strongly affect the current state of the memristors and preceding inputs diminish due to the short decaying time constant. The elements at the end of the sequence, thus, dominate the outcome and the memristor for the last input "A" has the largest output, as shown in Fig. 4d. For the middle-term case, memristors for the frequently appearing elements "C" and "D" make larger output values than that of the others, but "C" makes a larger value than "D" as "D" is mainly located at the beginning of the sequence compared to "C" (see Fig. 4e). Therefore, if the multiple W/I ratios are utilized for the memristors, different outputs will be obtained from the memristors with different W/I ratios for the same input, which can improve the training of the readout function. As shown in Supplementary Fig. 18, if the change of W/I ratio is larger than a minimum required time, the change in potentiation ratio and decaying time constant can be clearly observed.

**Training and generation of neuro-memristive computing for sequential data.** AMPs are natural antibiotics that exist in living organisms and act as antimicrobials by bursting the epidermis of bacteria or eukaryotes (see Fig. 5a). AMPs have an important role in the immune system of several living organisms, and many studies have been conducted regarding their use for development of new drugs for various diseases such as SARS and cancer[30,31]. Since AMPs are composed of one short peptide, they have the advantage of being able to mass-produce if their amino-acid sequences are known. The amphipathicity in which hydrophilic groups and hydrophobic groups in a peptide coexist in balance makes the AMP burst the bacteria. The AMPs are aggregated and inserted into the bacterial epidermis (surface of the bacteria). After that, because the bacterial epidermis is hydrophobic while the environment is hydrophilic, the hydrophobic side of the AMP is attached to the bacterial epidermis. The hydrophilic side of AMPs are faced with each other, and then the aggregated AMPs finally form a pore structure across the bacterial epidermis. Because of the pore consisting of AMPs, the cytoplasm of the bacteria flows outside and the bacteria is killed. This cell death process induced by AMPs is called bacterial lysis. However, the AMP should have a special structure, which is formed through complex, high-dimensional basis

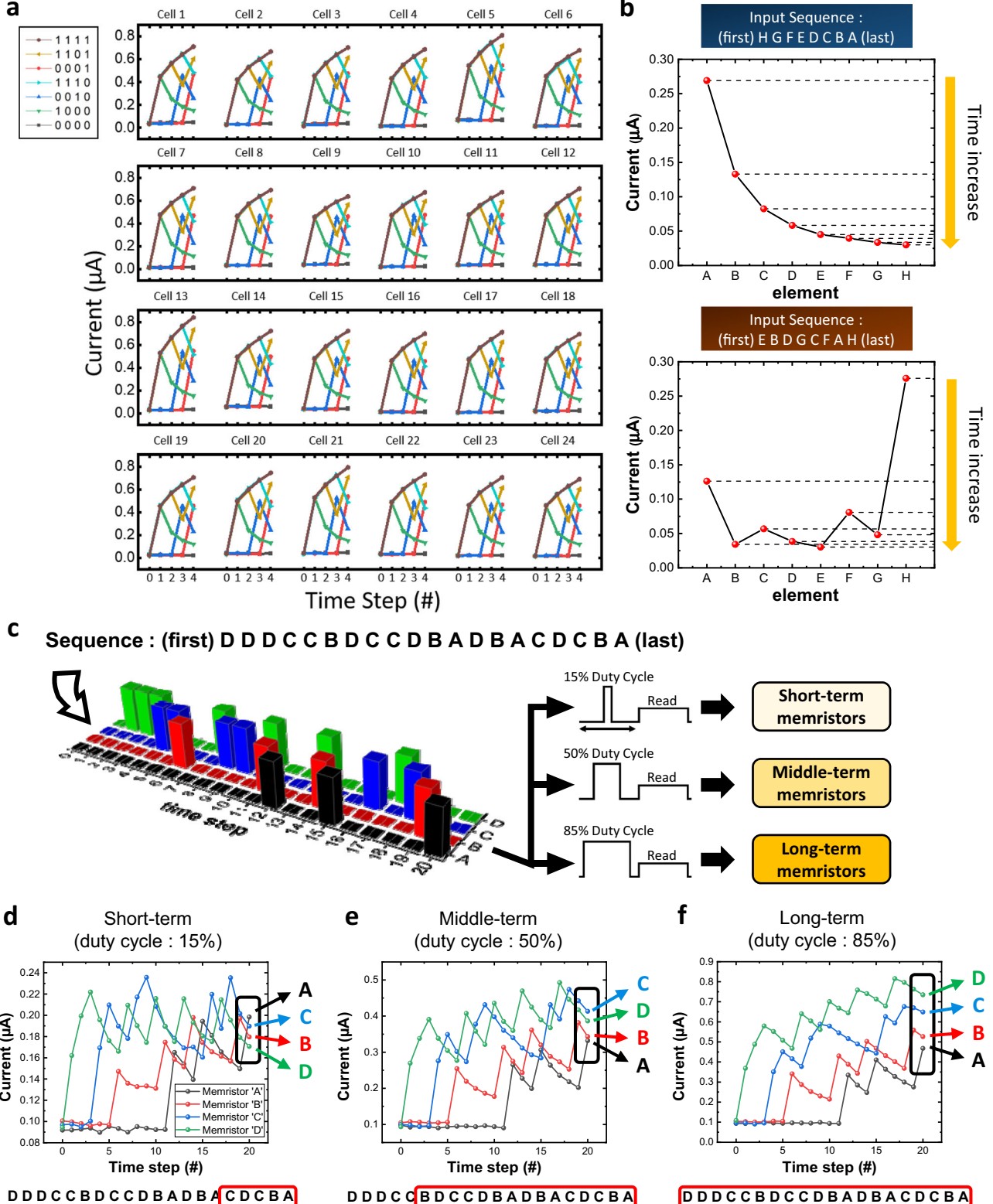

amino-acid grammar[32], as well as be amphipathic to interact with the bacterial epidermis and burst it. Therefore, it rarely makes the AMP by simply concatenating hydrophobic and hydrophilic short amino-acid sequences. In previous studies, it has been verified that LSTM-based neural networks could learn AMP's amino-acid grammar and create new potential AMP sequences[32,33]. However, LSTMs have too many trainable parameters and require too much complex computing to insert

the whole network into an edge processor. Instead of LSTM, we propose a neuro-memristive computing system based on the gradual TiO$_x$ memristor to generate new, potential AMP sequences.

To simulate the neuro-memristive computing based on the measured data from memristor arrays, we design an entire system on a circuit level (see Supplementary Fig. 19). The designed system's operation is conceptually represented in

**Fig. 4 Binary and sequence input processing of the gradual TiO$_x$ memristor. a** Response of 24 gradual TiO$_x$ memristors to the 4-bit inputs. The memristors are randomly selected from a 20×20 cross-bar array. Seven different binary inputs are used (brown: [1 1 1 1], yellow: [1 1 0 1], red: [0 0 0 1], sky blue: [1 1 1 0], blue: [0 0 1 0], green: [1 0 0 0], and black: [0 0 0 0]). For the input "1", the set pulse (4.3 V, 50 μs) is used while the ground is maintained for "0". The read pulse amplitude and width are 1.5 V and 200 μs, respectively. The gradual TiO$_x$ memristor successfully separates different binary inputs at the last time step (4), and all devices show similar responses without severe device-to-device variations. **b** Experimental results of memristors' output separating two sequences having different orders. Memristors for past elements have smaller output currents than memristors for recent elements.
**c** Schematic of the way how a sequence input (D D D C C B D C C D B A D B A C D C B A) is applied to memristors. The sequence is transformed into a pulse train and applied to the assigned memristors in order. The unit length of a single pulse (or ground) is 100 μs. The set pulse amplitude is 4 V, but only a portion (duty cycle) of the 100 μs is elevated to 4 V; the rest of the 100 μs is grounded. The read pulse (1.5 V, 100 μs) is applied between every set pulse and rest. **d–f** Memristors' output through the whole input sequence in a short-term, middle-term, and long-term memristors. The final results of each duty cycle case show the different processing abilities of memristors with different duty cycles. The memristors process sequences with a longer time domain along with an increase in the width/interval (W/I) ratio. The output of the long-term memristors only contains frequency information of each alphabet, while the order information is clearly shown in the short-term memristors. The sequence with the red box below the figure represents the range of the sequence that might affect the final output. All of the results in Fig. 4 are measured from a 20 × 20 cross-bar array.

Supplementary Fig. 20. Based on the design, the simulation of the neuro-memristive computing having four different W/I ratios (5/95, 10/90, 15/85, and 20/80) was tested for AMP generation.

First, the input sequence is transformed into a 20-dimensional (or 20-channel) vector having the same time step to the original sequence data. Each memristor in the cross-bar array processes the spike train from one channel (one amino-acid) among the 20 channels in the transformed input data (see Fig. 5b). In our system design, the system consists of four groups of memristors where each group has different W/I ratios to process the given input data differently (see Supplementary Figs. 19, 20, and "Methods"). To train the neuro-memristive computing system to generate new AMPs, the training data is pre-processed first. The target data to train the weights in the readout function is generated from the training data. For example, for the input sequence in Fig. 5b, the target data is made by removing the first amino-acid "G" and adding an end cursor "@" at the end of the sequence. Thus, the target data becomes "IGKFLHSAKKFG-KAFVGEIMNS@". The end cursor is used to inform the sequence end. The first amino-acid, which is "G", enters the memristor cross-bar array. Every column line represents each amino-acid used for AMP, and each row line represents a group of memristors having the same W/I ratio, as shown in Supplementary Figs. 19, 20. When the input amino-acid is "G", the column line for "G" is grounded and the other columns are floating. Then, the voltage pulses with various W/I ratios are transported from the row line and potentiate the memristors in column "G". After the potentiation of memristors for "G", a read stage is performed by applying the read voltage to the row lines while the column lines are grounded. The memristor cross-bar array output has 1 × 80 size because each group of memristors makes a 1 × 20 output and there are four groups. The cross-bar array output is an input of the readout function, and the readout function trains the weights based on the discrepancy between the readout function output and the corresponding target, which is the next amino-acid "I" in the given example. By repeating this process until the end cursor enters the system, the neuro-memristive computing system can learn the amino-acid grammar for the AMPs.

To compare the effect of using various W/I ratios, the four sequence sets were compared (a training set, generation sets from a short-term (low W/I ratio), long-term (high W/I ratio), and a mixed case(various W/I ratios)). The short-term, long-term, and the mixed case have the same readout function network size and the same number of trainable parameters to fairly compare the effect of mixed W/I ratios for data processing without the effect of network size. The result was evaluated by the generated output sequences' several features (probability of being AMP, global charge, molecular weights, and amino-acid distribution) using an AMP prediction tool[34] and a Python package for AMP analysis[35]. Figure 5c–f show the comparison among the training set and generation sets from short-term, long-term, and mixed case. The AMP prediction results in Fig. 5c show that 74.07% and 79.21% of sequences in a generation set from the short-term and long-term case, respectively, are predicted as AMP. Then, 84.11% of sequences from mixed case are predicted as AMP. All probability values of AMP in each set are represented in Supplementary Fig. 21. The average global charge of peptides representing the overall charge of amino-acid residues in each sequence[36] of the generation set from the mixed case is 2.32 pK$_a$ while the sequences from the short-term and long-term cases show 3.89 pK$_a$ and 3.11 pK$_a$, respectively (see Fig. 5d). The average global charge value from the mixed case is similar to the training set (2.28 pK$_a$), which means the neuro-memristive computing system with mixed W/I ratios can learn the complex pattern of the given data set. The molecular weights and the amino-acid distribution of the generated sequences from the mixed case are also similar to the training set, as shown in Fig. 5e, f. The different dynamic responses of memristors for different W/I ratios help the neuro-memristive computing system learn the complex amino-acid grammar and how to construct amphipathic peptides by transforming the input data into the higher dimension basis and making the training of the readout function effective.

Finally, the 3D peptide structures of an AMP sequence in the training set and a generated sequence with the highest probability of being AMP are shown in Fig. 5g, h. A real AMP and the generated sequence from our neuro-memristive computing system both show the similar structure and composition of hydrophobicity and hydrophilicity, which is one of the key points of the AMP. In summary, our neuro-memristive computing system composed of a gradual TiO$_x$ memristor cross-bar array demonstrates the potential of memristor-based computing for sequence type data by successfully generating new AMPs with a small number of parameters (only 50,520 parameters). All of these results reveal that the gradual TiO$_x$ memristor-based neuro-memristive computing system has great benefits regarding sequence type input data.

## Discussion

In summary, we developed a 20 × 20 high-density memristor array by utilizing gradual oxygen concentration metal oxide, which possesses high on/off ratio, excellent temporal/spatial uniformity, self-rectification, forming-free property, compliance

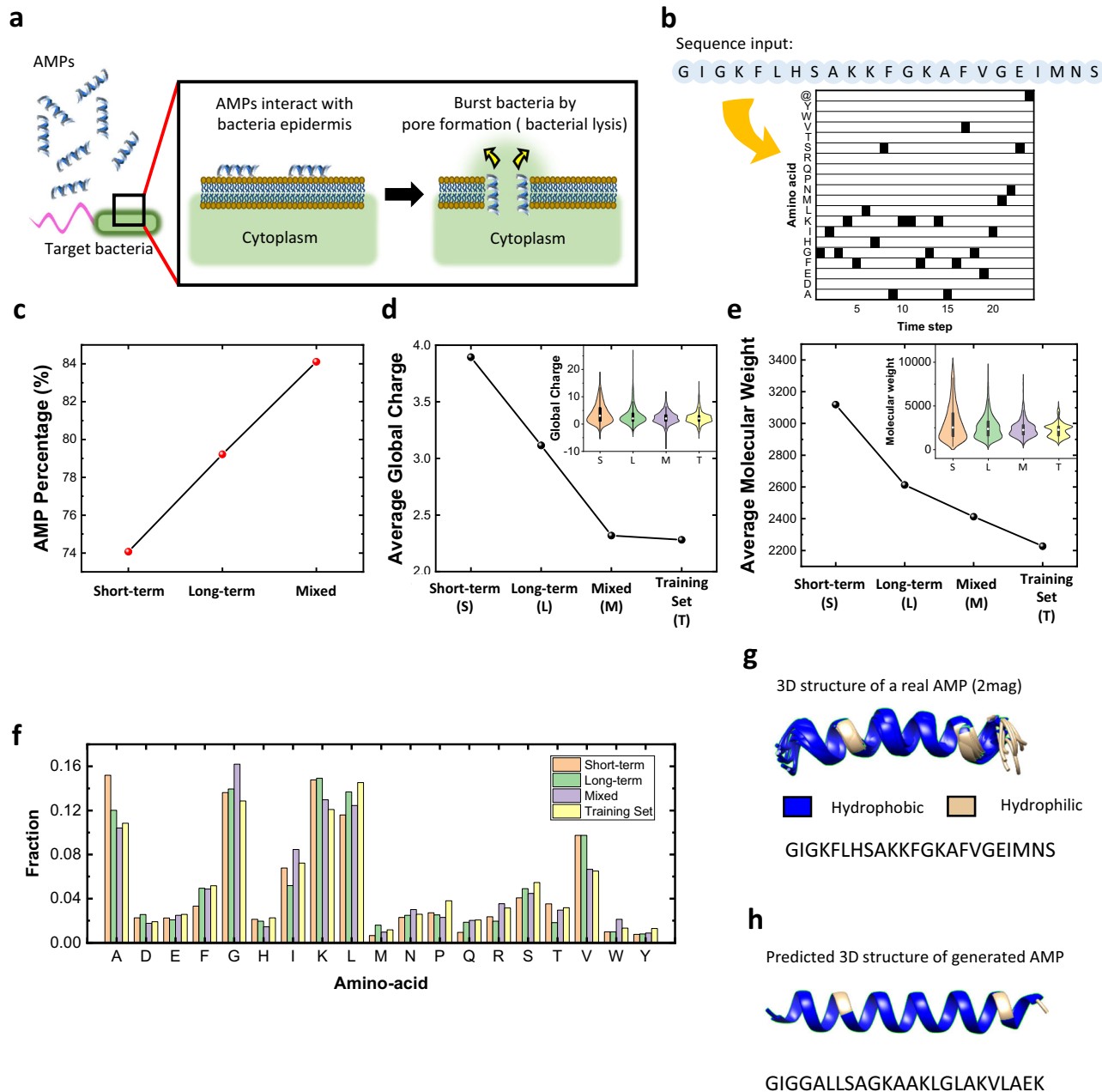

**Fig. 5 Schematic of the antimicrobial operation mechanism of AMP and the evaluation of generated AMP sequences. a** AMPs bursting the bacteria epidermis by pore formation. **b** An illustration of the transformation of an input sequence into a 20-dimensional vector (or 20 channels) with 24-time steps, where 20 is the number of the unique amino-acids in the input sequence with the added end cursor "@", and 24 is the length of the sequence after adding the end cursor. **c** The percentage of AMP sequences in the generation set from short-term (low W/I ratio), long-term (high W/I ratio), and mixed case (various W/I ratios), respectively. **d** Average global charge from each sequence set. The mixed case shows a similar global charge value to the training set, while the others do not. Inset: the total global charge value of the entire sequences in each set. **e** Average molecular weight from each sequence set. The mixed case generates sequences that have similar weights to the training set. Inset: the total molecular weights of all of the sequences. **f** Amino-acid distribution of each sequence set. **g** The 3D image of an AMP (protein data bank ID: 2mag[44]) in the training set. **h** A predicted 3D image of the generated AMP from the mixed case neuro-memristive computing system.

current-free property, and high yield for a memristor-based neuromorphic system. Unlike the other existing memristors operating with conductive filaments, our gradual TiO$_x$ memristor switches its resistive state through oxygen migration without strong filament formation and changes the effective insulator thickness. It showed dynamic analog operations, such as potentiation (set), depression (reset), and self-decaying. The newly developed memristor is utilized for memristor-based LIF

neuron, and shows several superior characteristics such as high uniformity, tunability, and low operation current (see Supplementary Table 1). It is demonstrated that the gradual TiO$_x$ memristor-based artificial neuron has similar features to the biological LIF neuron by modifying the presynaptic spike or the capacitance of the artificial neuron. Based on the reliable characteristics of the gradual TiO$_x$ memristor-based neuron, our results open up a path to developing a reliable

and energy-efficient neuromorphic processor with biological learning rules.

Furthermore, the gradual TiO$_x$ memristor cross-bar array as a short-term, leaky-integrate neuron is used to create a hardware-based neuro-memristive computing system. The newly developed system learns the AMP sequences and generates new AMP candidates. To improve the performance, different pulse width/interval ratios are utilized to effectively process the input. Neuro-memristive computing system with the gradual TiO$_x$ memristor having excellent uniformity could be used for mass-production because the trained readout function will have weights that are matched to the memristors in every single chip[12,37]. Because of the reduced training cost by substituting the recurrent layer of the LSTM into the memristors, the system can be suitable for online learning with high accuracy. The comparisons with the memristors in the previous works for similar neuro-memristive computing systems are shown in Supplementary Table 2. In addition, several superior properties, including self-rectification, forming-free, low-temperature fabrication, and high on/off ratio make the gradual TiO$_x$ memristor suitable for neuro-memristive computing system.

The proposed highly reliable memristor can be broadly used in neuromorphic processors that handle bio-plausible neural networks. Further improvements such as peripheral control circuits or non-volatile memristor arrays as synapses can be used for several practical applications including image classifications, speech recognitions, or real-time diagnosis with effectively reduced energy consumption and device size.

## Methods

**Memristor cross-bar array fabrication**. The newly developed memristor used in this work has a simple Pt/gradual TiO$_x$/Ti structure, as illustrated in Fig. 1a. The lithography method is utilized for the bottom electrode pattern, and the width of the electrode is 5 µm. The Ti 100 nm layer is deposited by an e-beam evaporator. After the bottom electrode deposition, the remaining photoresist is removed via the lift-off process. Through the second lithography step, a pattern is formed on the sample to cut off the BE-to-BE connections (Ti/TiO$_x$/Ti), which might induce severe leakage current in the cross-bar array. The Ti 12 nm layer, which will undergo the anodizing process, is deposited with an e-beam evaporator and is selectively removed via the lift-off process. The sample then undergoes the anodizing process to form the TiO$_x$ layer with gradual oxygen concentration by oxidizing the Ti 12 nm layer. The electrolyte used for the anodizing process is a NaOH 0.05 M, NH$_4$F 0.05 M mixed solution (total 0.1 M of NaOH and NH$_4$F in H$_2$O). The Ti 12 nm deposited sample is connected to the power supply and is soaked in the electrolyte. The anodizing process is done under the following conditions: anodizing voltage = 10 V, temperature = 20 °C, and anodizing time = 1 min. After the anodizing process, the top electrode pattern is formed through the lithography method, and the Pt 100 nm layer is deposited by the e-beam evaporator. Finally, the reactive ion etching method is used to remove all the exposed TiO$_x$; every leakage path is then removed. The whole structure of the fabricated 20 × 20 gradual TiO$_x$ memristor cross-bar array is well described in Supplementary Fig. 22. The anodizing process is used because it easily forms metal oxides with oxygen gradient, but other deposition methods including atomic layer deposition (ALD) or reactive sputtering could be used for fabrication of wafer-scale gradual TiO$_x$ memristor devices.

**Electrical measurement**. To measure the I–V curve of the gradual TiO$_x$ memristor, Keithley 4200A-SCS, a high-performance parameter analyzer, is utilized. The measurement system, including source measure units (SMUs) can precisely supply DC linear voltage sweep and simultaneously measure the output current.

The pulse responses of the memristor are measured by a data acquisition tool (National Instruments USB-6363) and a current preamplifier (DL Instruments Model 1211). The data acquisition tool (DAQ) generates pulse trains and simultaneously measures the output voltage, which is amplified by the preamplifier. The following electrical experiments (the 4-bit data processing experiment and the sequence data processing experiment) are conducted with the DAQ and the preamplifier measurement system.

For the 4-bit data processing experiment, the 4-bit data (ex. 1010) is transformed into the pulse train, which consists of set pulses (4.3 V, 50 µs) and ground (0 V, 100 µs). The pulse train is applied to the memristors in a 20 × 20 cross-bar array. After each bit ("1" or "0") is applied to the memristor, the read pulse (1.5 V, 200 µs) is applied to the memristor to obtain the conductance.

For the sequence data processing experiment, the sequence ([first] D D D C C B D C C D B A D B A C D C B A [last]) is transformed into the pulse train, which consists of set pulses (4 V) and ground (0 V). The unit length of each set and ground is 100 µs, but the 4 V is only applied as the amount of duty cycle value × 100 µs, while 0 V is maintained during the rest of the 100 µs. For example, if the duty cycle is 20%, the set pulse consists of 40 µs of the first ground, 20 µs of the 4 V set, and 40 µs of the last ground. The W/I ratio, in this case, is 20/80.

**Neuro-memristive computing simulation for AMP generation**. The training set consists of 1554 AMP sequences, which are collected from three publicly opened databases: the database of Anuran defense peptides (DADP)[38], the antimicrobial peptide database (APD)[39], and the a database for antimicrobial peptides (ADAM)[40]. Each AMP sequence in the data set is composed of ~7–48 amino-acid residues. All of the data are pre-processed by putting an end cursor at the end of each sequence. The end cursor is used to represent the location for the end of the sequence.

The simulation of the neuro-memristive computing system is done with realistic models of the memristor's pulsed responses for each W/I ratio. The results of the modeling are shown in Supplementary Fig. 23. Based on the realistic modeling of the memristor, the neuro-memristive computing system for the AMP generation simulator is constructed with four groups of memristors sharing same W/I ratio, where a single group consists of 20 memristors in a cross-bar array (see Supplementary Fig. 24). To clarify the effect of using mixed (various) W/I ratios, we also tested the neuro-memristive computing system with a single W/I case, while maintaining the readout function size. For mixed case, four different W/I ratios (5/95, 10/90, 15/85, and 20/80 among the total of 4 V and 100 µs set pulse) are used. The memristors with different W/I ratios process the input data differently especially in terms of the time window; thus, the readout function easily learns the pattern in the data set. Furthermore, short-term (low W/I ratio only) and long-term (high W/I ratio only) cases are examined, respectively, to observe the effect of the W/I ratio on the system performance. In the single W/I ratio cases, four groups of memristors share the same W/I ratio (5/95 for short-term and 20/80 for long-term) and the number of trainable parameters is equal to the mixed case.

In the training phase, the input data (AMP sequences) are fed into the memristor cross-bar array. Each amino acid in the sequence is one-hot encoded and is assigned to memristors; four memristors having different W/I ratios are used for a single amino-acid. The neuro-memristive computing system simulator learns AMP grammar by predicting the next amino-acid. If there is a sequence, "KAIST", for example, it is pre-processed with the end cursor "@" by attaching the end cursor to the end of the sequence. Now, the sequence is "KAIST@" and "K", "A", "I", "S", and "T" are the training data, and "A", "I", "S", "T", and "@" are the corresponding targets. In this example, "K" enters the system first, and a memristor for "K" is potentiated while the others remain in their initial states. The outputs from the memristors are then transmitted into the readout function. To efficiently train the readout function by using the outputs of all four W/I ratios, the outputs from each memristor group are differently amplified so the maximum output value of each memristor group becomes similar. The readout function is a simple 80 × 500 × 20 multi-layer perceptron (MLP) where the input dimension 80 comes from the number of memristors in the group, and the output dimension 20 comes from the number of selectable elements (19 amino-acids and 1 end cursor). The number of trainable parameters of the whole system is 50,520 for both single and mixed W/I ratio cases. The first and second layers have rectified linear unit (ReLU) as an activation function for fast and efficient learning, and the third layer has a softmax to make the output values in the form of probability. The system picks the next element based on the readout function output (see Supplementary Fig. 25). Then, based on the error between the corresponding target "A" and the readout function output, the readout function trains its weights by using a simple machine learning algorithm (logarithmic regression). The Python toolkit Keras is used to access TensorFlow. The same process (feed the input to the memristor cross-bar array, feed the memristor output to the readout function, pick the next output, and train the weight based on the error) is repeated until the training of a sequence is finished. When the end cursor becomes the corresponding target, the training of a sequence is finished, as there is no corresponding target for the end cursor. Then, the memristors are reset to the initial state. This whole process is done for every 1554 sequences. After the training iterates 50 times (50 epochs) with the whole training data, the training phase is finished and generation starts.

Generation of the AMP sequences is started by putting the first amino-acid (seed) in the memristor cross-bar array (see Supplementary Fig. 26). We used the first amino-acids of each AMP in the training data as a seed for convenience. The system predicts the next amino-acid for seed, and the selected amino-acid becomes the second amino-acid element of the generated sequence. The sum of the readout function output values is 1 due to the soft-max activation function. Thus, to give variation to the generated sequences, the next predicted amino-acid is selected based on the readout function output values as a probability distribution. The selected amino-acid is then fed into the memristor cross-bar array as an input, and these processes are repeated until the end cursor is selected. If the generated sequence already exists in the training data, then the sequence is removed from the generation set.

Finally, we generate 1554 sequences from each case (short-term, long-term, and mixed cases), and evaluate the properties of the sequences. CAMP, the evaluation tool for generated sequences, is an AMP prediction tool that presents the probability of the given sequence is AMP. In addition, the Python modlAMP package is used to analyze the generated sequences. By using this modlMAP package, the physical/chemical properties of the generated sequences, such as the global charges and molecular weights, could be calculated. Finally, PEP-FOLD 3, which provides a predicted 3D model of peptides, is used to visualize the generated sequences[41,42]. Through PEP-FOLD 3, the generated sequences are converted into a protein data bank (PDB) data format. We visualize this PDB data into the 3D model by using UCSF Chimera[43].

## Data availability
The data related to the figures and other findings of this study are available from the corresponding author upon reasonable request.

## Code availability
The code used for the simulation is available from the corresponding author with detailed explanations upon reasonable request.

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

## Acknowledgements

This work was supported by Samsung Research Funding & Incubation Center of Samsung Electronics under Project Number SRFC-IT2101-04.

## Author contributions

S.P. and S.C. conceived this work. S.P. and J.P. designed the device. S.P. designed the experiments and overall simulation. H.J. set up the electrical measurement platform and conducted measurement. S.P. and S.C. prepared the manuscript. All authors discussed and contributed the discussion and analysis of the results regarding the manuscript. S.C. supervised the study.

## Competing interests
The authors declare no competing interests.

**Additional information**

