## [Peer Review File · Nature Communications]

Title: Experimental demonstration of highly reliable dynamic memristor for artificial neuron and neuromorphic computingREVIEWER COMMENTS

Reviewer #1 (Remarks to the Author):

In this work, the authors fabricate a memristor with high endurance and low variability and claim that it is particularly suited to memristive reservoir computing (RC) systems. The proposed memristor has a decay mechanism or short-term memory which transforms spatio-temporal data to a high dimensional basis. Previous work has used similar memristor devices (with spontaneous ionic redistribution) for RC systems (reference 11), but the material and the device physics of this memristor could be leading to superior endurance and variability.

Given the novelty emphasis on being a memristive RC device, the paper should provide comparisons with other devices and demonstrate the improvements quantitatively to demonstrate the claims. Please see below for more detailed suggestions

The proposed memristor device demonstrates a novel ability to decay through the oxygen anions. This inherent self-decay feature can prove suitable in implementations of neurons for RC systems. In the case of an Echo State Network, the decay could act as the leakage factor, whereas for a Liquid State Machine the device could provide an inherent ability to act as the membrane potential of a spiking neuron. Even beyond RC methods, this device could be suitable for generic spiking neural network implementations. However, there are three key discussion points that I feel need to be addressed in this paper.

First, from my understanding this device cannot behave like traditional memristors where it can retain a specific resistance state but it is unclear as the authors state in line 147 " The LRS to HRS transition (reset) could occur without external stimulus due to the 148 diffusion of oxygen anions, which results in short-term memory effects with a 40 ms time constant." By saying it could occur, it is unclear if there is a probabilistic nature to the decay factor or if the device can become stuck at a LRS.

Second, because the novelty of this self-decay property is reliant on the W/I ratio, there should be a larger focus on the possibilities of the device. From the paper, the width/interval ratio is what determines the rate of decay. It would be beneficial to discuss the sizing of the devices for different W/I ratios, how/if device properties change, and minimum and maximum time-constants achievable.

Third, though the paper is framed for RC systems, it appears to be better suited towards an efficient neuron model using a memristor. The proposed memristors cannot serve as weights, but it is hard to assess the feasibility of designing neurons around the proposed device without the complete circuitry. The authors should focus on tailoring the discussion and simulations to show how this device can be realized to implement neuronal models, such as those in typical RC models, and how it compares in terms of efficiency with existing neuron models (memristor, analog, or digital).

RC models have less parameters than LSTMs and RNNs. This statement holds true for LSTMs which have complex gating models, but they have an equivalent number of parameters as RNNs. The only other

benefit for RC is the reduced training cost but from the discussion in the paper it does not focus on online learning. Rather they discuss a fixed, pre-trained readout which can be deployed to multiple crossbars due to the uniform nature of their fabrication.

Beyond that, at a high level the claim this as an RC computing model requires serious validation. From all the tests and architectures described there is no recurrent network. Rather, from the way it is presented, a set of memristors that act as leaky integrators are implemented and readout the final state of each individual memristor for classification. This is not a true RC model, and it is not clear how they treat the inputs to the network in terms of weights. In writing they discuss how the memristors see a combination of inputs, but in the alphabet example it is described as a 1-to-1 mapping where each memristor only sees a single input. A similar setup is described for the AMP sequences.

By framing the device contributions in the light of new neuron circuits and generalizable models, there could be lot more merit to this paper, rather than presenting it as an RC memristor.

Another concern is that the authors discuss the high-dimensional basis used for classification, but if every memristor is only a filtered version reflecting the activity of a single input category, each device belonging to the same input should be the same. This is almost like concatenating the input multiple times, but not truly expanding to a high-dimensional space where the input would become linearly separable. In this setup, one cannot see the need to use more devices than the number of unique input characters. Beyond this, how would the authors intend to extend the proposed approach to a single variable, continuous time-series data, or multivariable continuous data.

That being said, in terms of clarity I think the paper is poorly written in that it does not flow. It was very hard to follow the discussion and figures, and there seemed to be very little discussion around several key topics. First, the classification method relied on the decay of memristor states. If the decay is inherent to the memristor, it is unclear how it can be controlled to allow the memristor to hold a specific resistant state (the paper states that "decay from LRS to HRS could occur"). If it is not possible, there are limitations in how long information can be stored (likely dependent on the W/I ratio) but this is not thoroughly explored, and this device cannot be used as a synaptic weight which is common in memristor based neuromorphic circuits making it serve a very different purpose. If the decay can be controlled, what is the overhead associated with it? The other part that is difficult to understand is what the multi-modal RC setup is. It seems like it just refers to a system of crossbars with different W/I ratios which is confusing when used with the term multi-modal in ML. But then questions arise such as: how scalable is the approach with varying W/I intervals, how large are the devices, what are the limits in terms of ratios and their respective memory lengths.

Grammatical edits:

Line 27 - while using a small number of training

Line 60 - The mentioned bottlenecks can be reiterated and mapped to their solutions in the existing work in the later stages of the paper

Line 170 - anions across the whole TiOx bulk

Fig. 2 - (b) and (c) label spelling (electrode)

Reviewer #2 (Remarks to the Author):

The work by Park et al. presents a TiO_x-based memristor system and proposed an RC system based on such memristors. There are clearly interesting results, including the forming-free and uniform switching characteristics of the TiO_x memristors. The materials study also seems solid, and the topic is timely. However, several concerns need to be addressed.

RC systems are suitable for processing temporal data. It's thus puzzling why the authors picked the AMP task, which has no temporal components. In fact, the readout layer seems quite elaborate (multi-layer perception with nonlinear activations) already. Given the complex readout layer, instead of using an RC system, I would expect a conventional CNN network + the existing readout layer can probably achieve similar or better results.

The AMP task is achieved through simulation. The authors have enough memristors and have 100% yield, so why not implement it experimentally?

The experimental setup was very unclear. There was no discussion in the main text, and only scant information in "Methods". From Supplementary Information Figure S13, it seems the authors send 1 character to a corresponding memristor, instead of sending a sequence to 1 memristor. Why? Overall a careful justification for the experimental setup needs to be presented, and it should be included in the main text.

The authors highlight throughout the manuscript the idea of "gradual TiO_x" and showed the two regions of amorphous oxygen-rich layer and polycrystalline oxygen-deficient layer. Since the TiO_x layer is created through a single anodization of the Ti BE, why is the bottom layer polycrystalline while the top layer amorphous?

Several claims are over-reaching and incorrect. Including "To the best of our knowledge, this is the first study to demonstrate a large 1R memristor crossbar array with 100% device yield." (p6), and "For the first time, we trained and generated sequential data with memristive RC." (p15).

Reviewer #3 (Remarks to the Author):

This work combine two highly hot topics: memristors and machine learning. The main novelty of the work consist in application of memristors' array, while individual memristors possess average characteristics in comparison with other recently published based on TiO₂ (10.1063/1.5037835, 10.3389/fchem.2020.00724,10.1063/1.4940361) and other materials (10.1038/s41467-021-25455-0,10.1038/s41586-021-03748-0). Authors demonstrates important results and this study hs significant contribution to the field of application of memristors in machine learning. I recommend to publish this

article in Nature Communications after minor revision. Several questions are listed below

1) how was obtained fig. 1f,g and fig. S3? Is this photo or schemes?

I recommend also provide SEM of crossbar array or AFM

2) the choice of anodizing technique and obtained TiO_x layer should be explained in more detail. What is the advantage in comparison with PVD or ALD? What phase was obtained (rutile, anatase or others? Is this nanocrystals or nanotubes. This answers should be addressed, I recommend provide X-ray analysis for phase identification.

3) authors should be more accurate in discussing mechanism of TiO₂ and conclusions (p. 9, word file).

This paragraph sound as a guess

4) 20 is not very high on/off ratio. What is on/off ratio for individual memristor? Is 20 is average for all arrays? Fig. S4 provides only 2 cycles, for better demonstration of stability more cycles (at least 50, if authors claims 50 cycles at fig. 1 and 10⁶ cycles at p.6).

5) p. 13, clarification for mechanism of killing cell should be provided "The positively charged hydrophilic portion of the peptide attaches to the negatively charged surface of the bacteria, and the hydrophobic portion interacts with proteins on the surface of the bacterial cell to induce "bacterial lysis", which kills the cell"

Response Letter to Reviewers' Comments

We sincerely appreciate the valuable time the reviewers have spent reviewing our manuscript and providing insightful comments and suggestions to help further improve the quality of our work. Considering the reviewers' evaluations, we have made a point-by-point response to the reviewers' comments, and revised our manuscript to improve the clarity of our work. We believe we have addressed all of the reviewers' comments and now the paper is more rigorous in content and clearer in presentation. Based on the responses below, we have appended twelve figures in the revised manuscript, twelve figures and two tables in the supplementary information, to address the comments. Our point-by-point responses to the reviewers' comments are as follows.

Reviewer #1

In this work, the authors fabricate a memristor with high endurance and low variability and claim that it is particularly suited to memristive reservoir computing (RC) systems. The proposed memristor has a decay mechanism or short-term memory which transforms spatio-temporal data to a high dimensional basis. Previous work has used similar memristor devices (with spontaneous ionic redistribution) for RC systems (reference 11), but the material and the device physics of this memristor could be leading to superior endurance and variability.

Given the novelty emphasis on being a memristive RC device, the paper should provide comparisons with other devices and demonstrate the improvements quantitatively to demonstrate the claims. Please see below for more detailed suggestions

Response: We thank the reviewer for the constructive comments on our work and also for pointing out the important insight to emphasize the results. To reflect the reviewer's evaluation on the manuscript, we have revised the title of the manuscript, modified several parts of the main text, and added additional experiments and discussion to address pointed issues. We also have added the comparisons with other devices in terms of an artificial neuron device and a RC device. Our detailed responses to your comments are provided below.

Comment #1:

The proposed memristor device demonstrates a novel ability to decay through the oxygen anions. This inherent self-decay feature can prove suitable in implementations of neurons for RC systems. In the case of an Echo State Network, the decay could act as the leakage factor, whereas for a Liquid State Machine the device could provide an inherent ability to act as the membrane potential of a spiking neuron. Even beyond RC methods, this device could be suitable for generic spiking neural network implementations. However, there are three key discussion points that I feel need to be addressed in this paper.

Response: Thank you very much for this insightful comment. As the reviewer mentioned, the device properties are not suitable only for the RC system, but also for the generic electrical neuron. Therefore, we have conducted additional experiments for the developed memristor based artificial neuron. We have added additional experimental data and discussion about the memristor based artificial neuron.

To emphasize the device operation as an artificial neuron, we have changed the manuscript title to “**Experimental demonstration of highly reliable dynamic memristor for reliable artificial neuron and neuromorphic computing**” from “Experimental demonstration of highly reliable dynamic memristor array for reservoir computing”. In addition, we have added a subsection “Reliable LIF neuron with gradual TiO_x memristor” to discuss the neuron operation of the device. We also have revised the abstract, introduction, and conclusion sections to reflect the results and discussion about the neuron operation.

Detailed explanations are provided below.

Comment #2:

First, from my understanding this device cannot behave like traditional memristors where it can retain a specific resistance state but it is unclear as the authors state in line 147 " The LRS to HRS transition (reset) could occur without external stimulus due to the 148 diffusion of oxygen anions, which results in short-term memory effects with a 40 ms time constant." By saying it could occur, it is unclear if there is a probabilistic nature to the decay factor or if the device can become stuck at a LRS.

Response: Thank you for this valuable comment. We agree that our original statement is ambiguous and may be misleading. The device in this work is volatile, which means that it has short-term retention instead of long-term retention. Therefore, to inform this, we removed the word “could” from the original sentence and added the word “volatile” to emphasize the device has a short-term memory effect as follows:

Page 8, line 166: “**The LRS to HRS transition (reset) occurs without external stimulus due to the diffusion of oxygen anions, which results in short-term (volatile) memory effects with about 40 ms time constant.**”

Comment #3:

Second, because the novelty of this self-decay property is reliant on the W/I ratio, there should be a larger focus on the possibilities of the device. From the paper, the width/interval ratio is what determines the rate of decay. It would be beneficial to discuss the sizing of the devices for different W/I ratios, how/if device properties change, and minimum and maximum time-constants achievable.

Response: Thank you very much for the constructive comments. As the reviewer suggested, we added more explanations for the device and conducted experiments as shown below.

(1) The effect of the device sizes and pulse width/pulse interval (W/I) ratios on device properties.

First, we have conducted the device measurements with various device sizes and W/I ratios and verified that the device size changes the current magnitude while the device properties such as potentiation/decaying characteristics remain same. The devices with different sizes (5×5 , 10×10 , 20×20 , $50\times 50\ \mu\text{m}^2$) are investigated with various pulse conditions, as shown in Figure R1. The devices are potentiated with consecutive 100 set pulses (4 V) with various W/I ratios (10/90~30/70) within 100 μs . As shown in the Figure R1, the device current reduces as the device size decreases due to the non-filamentary nature of the device. Moreover, it does not significantly change the other properties such as decaying speed and potentiation ratio for each W/I ratio, which means the device is scalable without property degradation. For example, the decaying time constant is about 40 ms for the largest W/I ratio case (30/70) for every device regardless of the device size. To clarify these, we appended sentences:

Page 9 and line 176: “In Supplementary Fig. 7 the device current reduces as the device size decreases due to the non-filamentary nature of the device. Moreover, it does not significantly change the other properties such as decaying speed and potentiation ratio, which means the device is scalable without property degradation.”

The corresponding Figure R1 is also appended in the **Supplementary Figure 7** in the Supplementary information.

Figure R1. The response of devices with different device sizes and W/I ratios. Four devices having different sizes from $5 \times 5 \mu\text{m}^2$ to $50 \times 50 \mu\text{m}^2$ are measured by the read pulses (1.8 V, 400 μs) during the consecutive 100 set pulses (4 V) and following decaying. The pulse widths are varied from 10~30 μs and the corresponding pulse intervals are 90~70 μs . The responses from each device are identical except current scaling effect due to device size difference.

(2) The maximum and minimum of time constants

To investigate the minimum and maximum decaying time constant of the device, the gradual TiO_x memristor is potentiated with various W/I ratios from 5/95 to 50/50 within 100 μs , and the decaying time constants are analyzed. As shown in the Figure R2, the decaying characteristic is changed when the W/I ratio is changed. When the W/I ratio is large, the longer set pulse width results in a larger amount of the oxygen anions movement into the Pt-gradual oxide interface. This causes slow decaying speeds and a long time constant. The maximum time constant (~38 ms) is achieved when the W/I ratio is larger than 40/60. The minimum time constant (~15 ms) is achieved when the W/I ratio is smaller than 10/90. The decaying time constant can be adjusted by modifying the W/I ratio and we added the following sentences in the main text:

Page 17 and line 357: “As shown in the Supplementary Fig. 17, a high pulse width/interval (W/I) ratio makes the decaying speed slower and increases the time constant, while a low pulse W/I ratio makes the decaying speed faster and decreases the time constant. The decaying time constant can be controlled from 15 ms to 40 ms by adjusting the W/I ratio, and the different decaying speed enables the memristors to process the same input data differently.”

We also appended Figure R2 as a Supplementary Figure 17, to show the time constant change along with the different W/I ratio.

Figure R2. The normalized responses of the gradual TiO_x memristor during the potentiation and decaying with various W/I ratios.

Comment #4:

Third, though the paper is framed for RC systems, it appears to be better suited towards an efficient neuron model using a memristor. The proposed memristors cannot serve as weights, but it is hard to assess the feasibility of designing neurons around the proposed device without the complete circuitry.

The authors should focus on tailoring the discussion and simulations to show how this device can be realized to implement neuronal models, such as those in typical RC models, and how it compares in terms of efficiency with existing neuron models (memristor, analog, or digital).

Response : Thank you for the insightful comments and suggestions about the experiments. We agree that our work can be implemented as an artificial neuron device using short-term memory effect such as a leaky-integrate and fire (LIF) neuron and a short-term neuron for RC. Therefore, we have conducted several experiments to prove the proposed memristor can be used as a reliable artificial neuron. We also discussed how the developed memristor can be realized as a LIF neuron. In addition, we appended the two comparisons; 1) LIF neuron devices, 2) RC devices in terms of efficiency. To see the feasibility of the memristor based RC, we have appended a designed circuitry for the proposed RC system. The detailed explanations are provided below.

To focus on the device property as an artificial neuron, we have conducted experiments about the memristor based LIF neuron and added a new subsection “**Reliable LIF neuron with gradual TiO_x memristor**” in the main text to explain the results for a highly reliable artificial neuron, and tailored the discussion. The subsection includes (1) a comparison between the biological neuron and the memristor based artificial neuron, (2) a complete circuitry for the artificial

neuron with the proposed memristor, and (3) the superior characteristics of the artificial neuron based on the gradual TiO_x memristor, especially in terms of the uniformity.

(1) Short-term plasticity and LIF neuron operation of the gradual TiO_x memristor based neuron.

First, to show that the device can operate as a highly reliable neuron device, PPF (paired-pulse facilitation) and PPD (paired-pulse depression), which are the short-term plasticity of the biological neuron, are measured with two different input pulse train frequencies to the device. The input pulse has a 3 V amplitude and a 100 μs duration. When the pulse train frequency is high (3,333 Hz), the device conductance increases, which is similar to the short-term potentiation in the biological neuron (PPF) (see Figure R3). On the other hand, if the pulse train frequency is low (625 Hz), the conductance decreases, which is similar to the PPD of the biological neuron. These short-term plasticity features can be used as a leaky-integrator. The short-term plasticity is also applicable to the reservoir computing system, that requires the leaky-integrate neurons in the reservoir.

Second, to emulate a biological LIF neuron, we built a simple artificial LIF neuron based on the memristor device. The circuit schematic of the artificial neuron and the comparison to the biological neuron are shown in Figure R4. The LIF neuron circuit based on the volatile memristor is widely studied in previous studies (e.g. Wang, Z. et al. [*Nat. Electron.*, 2018], Kumar, S. et al. [*Nature*, 2020], Duan, Q. et al. [*Nat. Commun.*, 2020]). As represented in Figure R4, the parallelly connected capacitor and the memristor are analogous to the biological neuron's soma, that integrates the signals from the presynaptic neurons and makes an action potential when it fires. The serially connected non-volatile memristor represents the synaptic weight between the neuron and the presynaptic neuron, where the small resistance means the strong connection between them. In this study, to exclude the non-idealities from the non-volatile memristor for the analysis of the artificial neuron, we substituted the non-volatile memristor to a general resistor.

Figure R3. The short-term plasticity of the gradual TiO_x memristor along with the different input spike frequencies. The gradual TiO_x memristor is tested with the input pulse train (3 V, 100 μs) having 3,333 Hz (high frequency) followed by 625 Hz (low frequency). When the input pulse train has a high frequency, the device conductance increases, which is analogous to the PPF (paired-pulse facilitation) in a biological neuron. When the input pulse has a low frequency, the device conductance decreases, which is analogous to the PPD (paired-pulse depression) in a biological neuron. The device conductance decreases when the low frequency pulse train is applied, even though the input pulse amplitude remains the same, because of the short-term memory effect of the gradual TiO_x memristor.

Figure R4. An illustration of the operation of a biological neuron (left) and the developed memristor-based artificial neuron (right). The synapse between the presynaptic neuron and the post-synaptic neuron, the soma which integrates the presynaptic spikes, and the post-synaptic spikes of the biological neuron are represented as a non-volatile memristor, a volatile memristor and parallelly connected capacitor, and the spikes from the volatile memristor, respectively, in the artificial neuron.

Based on the artificial neuron composed of a $47\text{ k}\Omega$ serial resistor, a 10 nF capacitor and the proposed memristor, we have measured LIF neuron characteristics and analyzed the results (see Figure R5). First, when the 4 V , $200\text{ }\mu\text{s}$ input spike train with a $50\text{ }\mu\text{s}$ interval is applied to the device, the artificial neuron generates post-synaptic spikes after integrating two pre-synaptic spikes. During the first two input spikes, the artificial neuron does not make output spikes but just integrates the charges in the capacitor. After the two spikes, the artificial neuron's potential (or the capacitor potential) exceeds the threshold voltage of the memristor. Then, the memristor is turned into the low resistance state and conduct current, which is an analogy of the firing of a biological neuron.

Figure R5. The LIF neuron operation of the developed memristor-based artificial neuron, and the used circuitry. The gradual TiO_x memristor is connected with a serial resistor and a parallel capacitor, which serve as a synapse and a soma, respectively. At first, the memristor-based neuron does not fire because the input spike is accumulated in the capacitor. When the capacitor is charged and the potential across the capacitor becomes large as much as the set voltage (threshold voltage) of the memristor, the memristor is triggered to the low resistance state and the neuron fires. In the given resistance and capacitance, the artificial neuron requires two input spike pulses (presynaptic spikes) to exceed the threshold in this figure, as shown in the output current (red line) of the neuron.

(2) Characteristics of the developed artificial neuron

To test the characteristics of the gradual TiO_x memristor-based artificial neuron, we tested the artificial neuron's responses according to (1) the different presynaptic spike frequency, (2) synaptic strength (serial resistor's conductivity), (3) presynaptic spike amplitude, (4) different capacitor, and (5) the different resistor and capacitor with the same $R \times C$ value. (Figure R6-10).

As shown in Figure R6, when the presynaptic spike frequency is low (1,600 Hz), the artificial neuron does not generate post-synaptic spikes, because the integrated charges in the capacitor are depleted during the interval between spikes. However, when the input frequency increases, the artificial neuron generates post-synaptic spikes. This frequency dependent firing characteristic is important to learning in the human brain, by potentiating (depressing) synapses between frequently (infrequently) firing presynaptic neurons and a post-synaptic neurons.

Figure R6. The responses of the developed memristor-based artificial neuron along with various presynaptic spike frequencies. The artificial neuron is tested with the presynaptic spike trains (200 μ s, 4 V) with various frequencies (1,600, 2,400, 3,200, and 4,000 Hz) to investigate the artificial neuron responses. When the frequency is 1,600 Hz (low), the artificial neuron does not fire, even though the presynaptic spike trains are applied. For the frequencies larger than 2,400 Hz (high), the artificial neuron fires. The used artificial neuron is composed of the gradual TiO_x memristor, a 47 $\text{k}\Omega$ resistor and a 10 nF capacitor.

Figure R7 shows the different responses of the artificial neuron with different serial resistor (94 $\text{k}\Omega$ for left plot and 23.5 $\text{k}\Omega$ for right plot). The synapse between two neurons can be represented as a resistor. When the resistance is high (94 $\text{k}\Omega$), the synapse is weak and the neuron fires after integrating five presynaptic spikes. On the other hand, when the resistance is low (23.5 $\text{k}\Omega$), the synapse is strong and the neuron fires within integrating a single spike. The result for moderately connected synapse (47 $\text{k}\Omega$) is shown in Figure R5, and it integrates two spikes for firing. Biological neurons that are connected with strong synapses fire together because the spikes from the strongly connected presynaptic neuron easily induces firing of the post-synaptic neuron. If the two neurons are fired simultaneously, they are wired together (potentiation of the synapse). This means that the firing of the post-synaptic neuron right after the firing of the pre-synaptic neuron makes the synapse stronger. This is called “Hebbian learning”, and is considered a fundamental learning mechanism of our brain. The developed artificial neuron in Figure R7 also shows the same property to the biological neuron; the required number of spikes for post-synaptic neuron firing decreases when the synapse is strong. Therefore, these results demonstrate that the developed artificial neuron can be utilized for the neuromorphic processor with biological learning rules.

Figure R7. The effect of the synaptic strength to the artificial neuron response.

Figure R8 shows the responses of the artificial neuron with different presynaptic spike amplitudes (2.5, 3.5, 4, and 4.5 V). When the spike amplitude is low (2.5 V), the artificial neuron does not fire. This is because the charges integrated in the capacitor are too low to generate high voltage for turning on the memristor. However, if the input amplitude is higher than 3.5 V, the neuron starts firing after integrating one or two presynaptic spikes. This result shows the artificial neuron can be modulated by the spike amplitude.

Figure R8. The effect of the presynaptic spike amplitude to the artificial neuron response.

The developed artificial neuron's characteristic can be easily modified by changing the capacitor, which determines the threshold voltage of the neuron (see Figure R9). When the capacitance is small, the capacitor's potential, which analogous to the membrane potential of the biological neuron, is easily elevated up to the threshold voltage. Therefore, when a 5 nF capacitor is used in the artificial neuron, the neuron fires after integrating a single presynaptic spike. However, if the capacitance increases to 10 nF, it requires two pulses to fire the post-synaptic spikes. Finally, when the capacitance is 20 nF, the neuron requires five spikes to fire. These results demonstrate that our artificial neuron can be easily tuned by modifying the capacitor.

Figure R9. Modulation of the threshold characteristic of the artificial neuron by changing the capacitance.

As shown in the Figure R10, when two neurons have different resistance and capacitance but the RC value remains same, the neuron shows same responses. This is because the threshold voltage is determined by the charges in the capacitor, and the speed of charging the capacitor is represented as $R \times C$. In the experiments, an artificial neuron is composed of 94 k Ω resistance and 10 nF capacitance (a neuron that easily fires but has a weak synapse), and the other is composed of 47 k Ω resistance and 20 nF capacitance (a neuron that hardly fires but has a strong synapse). The neurons show similar response to the presynaptic spike trains.

Figure R10. The effect of the resistor and capacitor to the artificial neuron’s characteristic.

(3) Spatio-temporal variation

Further investigations about the device-to-device variation, cycle-to-cycle variation, and operation speed of the artificial neuron have been conducted, and the results are shown in the Figure R11-13.

To assess the device-to-device variation of the developed gradual TiO_x memristor based artificial neuron, randomly selected 22 devices are used for the artificial neuron and measured. In the figure R11, the results of the 22 neurons are plotted. The 22 neurons show identical responses, which means excellent device-to-device uniformity. It proves the superiority of the gradual TiO_x memristor for the artificial neuron device with high uniformity.

Figure R11. The LIF neuron operations from the randomly selected 22 devices. All of the 22 devices show similar response along the presynaptic spike train (200 μ s, 4.5 V). These

results demonstrate the superior device-to-device uniformity of the gradual TiO_x memristor based artificial neuron.

In figure R12, the cycle-to-cycle of the developed artificial neuron is assessed. The artificial neuron is tested with 50 cycles of pulse train, where a single pulse train consists of 4 sub-pulse train, and a sub-pulse train has 20 pulses (4 V, 100 μs , and 50 μs interval). For one cycle, four sub-pulse trains are applied to the neuron with 1 ms interval. During the test, the responses of the artificial neuron are not significantly changed, as shown in the figure R12. Because of high uniformity of the gradual TiO_x memristor-based artificial neuron, a neuromorphic processor with high reliability could be utilized with the proposed device.

Figure R12. The cycle-to-cycle uniformity of the artificial neuron.

(4) Operation speed of the artificial neuron

To test the operation speed of the artificial neuron, the artificial neuron has been activated in various presynaptic spike conditions, as shown in the Figure R13. The presynaptic spike width and interval decrease, while keeping the same ratio, from 400 μs to 20 μs for spike width and from 100 μs to 5 μs for spike interval. These results demonstrate that the neuron operates with short pulse as short as 40 μs of spike width.

Figure R13. The effects of the pulse (spike) width to the operation of the artificial neuron. Presynaptic spike trains with various pulse widths are applied to the artificial neuron to investigate the operation speed of the artificial neuron. The neuron operates with spikes as short as 40 µs. During the test, the ratio between spike width and spike interval keeps constant.

To inform the above additional contents in the manuscript, we have appended Figure R3-R7 and R11 in the manuscript as Figure 3a-3l, and others in Supplementary Figure 10-14 in Supplementary information. The Figure 3a-3l which is appended in the revised manuscript is shown in Figure R14.

Figure R14. Gradual TiO_x memristor-based LIF neuron. **a**, Illustration of a biological neuron and the analogous memristor-based artificial LIF neuron. **b**, The short-term plasticity of the gradual TiO_x memristor along with the different input spike frequencies. **c**, The LIF neuron operation of the developed memristor-based artificial neuron. Applied spike train condition: 4 V, 200 μs width, and 50 μs interval **d**, The schematic of the circuitry used for the gradual TiO_x memristor-based artificial neuron. **e-g**, The output of the artificial neuron along with different presynaptic spike frequencies (3,200 Hz (e) and 1,600 Hz (f)), and the illustration of the experimental setup. During the test, the pulse amplitude and width are fixed (4 V, 200 μs). **h-j**, The output of the artificial neuron along with the different synaptic strengths (different resistor). When the synaptic strength is weak (large resistance (h)), the artificial neuron fires within six spikes, while it requires fewer spikes when the synaptic strength is strong (small

resistance (i)). **k** and **l**, The outputs of the 22 different artificial neurons demonstrate low variation of the gradual TiO_x memristor-based artificial neuron.

We also added the newly appended subsection “Reliable LIF neuron with gradual TiO_x memristor” to reflect the correct implication.

Page 10-13:

“Reliable LIF neuron with gradual TiO_x memristor

A biological neuron receives spike signals from presynaptic neurons and integrates the signals into its membrane potential (see Fig. 3a). When a neuron receives spikes in a certain interval, the neuron’s membrane potential is elevated. Once the membrane potential exceeds the threshold potential, the neuron sends spikes (an action potential) to its post-synaptic neuron. However, because the neuron has leaky-integrate property, the membrane potential goes back to its original potential (a resting potential) if the presynaptic neuron stops sending spikes and the membrane potential does not reach the threshold. This leaky integrator property is represented as the short-term plasticity of a neuron and induces “paired-pulse facilitation (PPF)” and “paired-pulse depression (PPD)” which are the biological neuron’s short-term plasticity phenomena. The leaky-integrate and fire (LIF) neuron model elaborately describes these biological neuron’s properties¹.

To demonstrate the gradual TiO_x based memristor as an artificial neuron device, PPF and PPD characteristics of the device are tested with the different presynaptic spike frequencies as shown in Fig. 3b. The different pulse intervals (200 μs and 1500 μs) are applied to the 10×10 μm² gradual TiO_x memristor, while the input voltage amplitude and width (3 V, 100 μs) keep constant. The memristor conductance increases with high input pulse frequency (3,333 Hz), while it decreases with low frequency (625 Hz). This means that the short-term memory effect of the gradual TiO_x memristor induces a short-term potentiation and depression, which is analogous to the short-term plasticity of the biological neuron.

In addition to the short-term plasticity, the gradual TiO_x memristor-based LIF neuron is constructed with a parallel capacitor and a serial resistor (see Fig. 3c and 3d). The parallelly connected volatile memristor and capacitor operate as a LIF neuron’s soma, where the biological neuron integrates spikes and makes an action potential when the membrane potential reaches the threshold. The resistor (or non-volatile memristor) that serves as a synaptic weight is serially connected to the gradual TiO_x based volatile memristor and capacitor^{2,3}. To examine the neuron operation without non-ideality factors from the synapse device, a fixed resistor is utilized. The resistance is low if synaptic weight is high, and it is high if synaptic weight is low. If the parallel capacitor integrates enough charges from presynaptic spikes in a certain time interval, the capacitor voltage becomes high enough to potentiate the volatile memristor. For example, The LIF neuron operation of the gradual TiO_x memristor-based artificial neuron with a 47 kΩ resistor and a 10 nF capacitor is shown in Fig. 3c. After applying two presynaptic voltage pulses, the neuron reaches the threshold voltage and fires.

The artificial neuron’s characteristics in terms of input spike frequencies are investigated in Fig. 3e, 3f, and 3g. The two different frequency spike trains (3,200 Hz, and 1,600 Hz) are applied to the memristor-based artificial neuron. When the frequency is high, the neuron fires within three spikes. However, the neuron does not fire if the input spike frequency is low at

1,600 Hz. This is because the developed neuron has leaky-integration property, and the capacitor voltage does not reach the threshold voltage.

Furthermore, in a biological neuron, the neuron easily fires when it receives spikes from a strongly connected presynaptic neuron, while it requires much more spikes to fire when the presynaptic neuron is weakly connected. This biological phenomenon is emulated in our artificial neuron by modifying the synaptic weight from the serial resistor (see Fig. 3h, i, and j). When an artificial neuron is composed of a gradual TiO_x memristor, a serial resistor (47 k Ω), and a parallel capacitor (10 nF), three presynaptic spike pulses are needed to make the neuron fire (see Fig. 3c). If the serial resistance increases to 94 k Ω , that represents a weak connection to the presynaptic neuron, the artificial neuron needs six consecutive spikes to fire (see Fig. 3h). On the other hand, if the serial resistance decreases to 23.5 k Ω representing a strong synaptic connection, the artificial neuron fires within two spikes as shown in Fig. 3i. In a human's biological neural network, the synaptic weight is long-term potentiated when the post-synaptic neuron fires just after the presynaptic neuron fires, because it means the post-synaptic neuron's firing event is strongly related to the presynaptic neuron's firing. This biological phenomenon is called a "Hebbian learning rule", and this is considered as a basic learning mechanism of biological neural networks⁴. The results show that the memristor-based artificial neuron behaves as the biological LIF neuron, and therefore, can be utilized for reliable neuromorphic processors with biological learning rules.

To build a reliable neuromorphic processor with memristor-based neurons, a highly reliable neuron operation is required. In Figure 3k and 3l, the gradual TiO_x memristor-based artificial neurons show similar neuron responses, which means the proposed memristor based artificial neuron has low device-to-device variation. The identical spike trains are applied to the randomly selected 22 artificial neurons, and the outputs of the neurons are shown in Fig. 3k. Every neuron operates in the same manner, without any noticeable differences. Moreover, we also test the cycle-to-cycle uniformity by applying the same pulse train 50 times to a single artificial neuron (see Supplementary Fig. 10). The response of the artificial neuron is not significantly different, demonstrating its low cycle-to-cycle variation. The low variation of the artificial neuron originates from the reliable switching behavior of the gradual TiO_x memristor.

To further investigate the characteristics of the gradual TiO_x memristor-based artificial neuron, the artificial neuron's responses are measured in terms of presynaptic spike amplitude and presynaptic spike width (see Supplementary Fig. 11 and 12). As shown in Supplementary Fig. 11, the artificial neuron does not fire with a small amplitude of the presynaptic spikes (2.5 V), but it fires with a large amplitude of the spikes (>3.5 V). In Supplementary Fig. 12, it is demonstrated that the neuron can operate with a short pulse (40 μs). Moreover, the artificial neuron's responses with the different combinations of the circuit components are measured; (1) different capacitance values and (2) different capacitance and resistance values while keeping the same $R \times C$ value (see Supplementary Fig. 13 and 14). The developed artificial neuron has a great tunability to adjust the threshold, by changing the parallelly connected capacitor or the serially connected resistor.

The gradual TiO_x memristor-based neuron demonstrates superior device-to-device and cycle-to-cycle uniformity, tunability, low spike current (10~35 μA), high off-state resistance (~100

MΩ), and an acceptable speed (>40 μs). The comparison table to the various memristor-based artificial neurons is shown in Supplementary Table 1.”

(5) Comparisons to the existing devices as an artificial neuron

Furthermore, as the reviewer suggested, we provide comparisons with other devices as LIF neurons to demonstrate the improvements. We compare the gradual TiO_x memristor to other memristors in terms of the artificial neuron in Table R1. As shown in Table R1, the diffusive memristor-based neurons are fast and have high off-state resistances, which are great for energy efficiency by preventing the leakage current, but they have significant uniformity problems because of the stochastic nature of the ionic movement. The Mott transition type memristor-based neurons are the fastest memristor-based neurons. Therefore, they have been used to imitate several characteristics of biological neurons. However, the low off resistance of the Mott transition memristors causes power consumption. To realize a reliable neuromorphic processor, the artificial neuron should have properties such as acceptable speed, great uniformity, high endurance, low current consumption, and high off resistance.

Device structure	Operation mechanism	Cycle-to-cycle variation	Device-to-device variation	Endurance	Spike peak current (μA)	Power at spike peak (μW)	Off resistance (R _{off}) (MΩ)
Pt/gradual oxygen concentration TiO _x /Ti	Ionic	1.39%	3.87%	> 5×10 ⁶	10~35	35	100
Pt/SiO _x N _y :Ag/Pt [R5, R6]	Ionic (diffusion of Ag)	N/A	N/A	N/A	20	10	200
Ag/SiO ₂ /Au [R7]	Ionic (diffusion of Ag)	N/A	N/A	N/A	23	0.5	10,000
Pt/TiN/NbO ₂ /TiN/W [R8]	Mott	Good, but not quantitatively analyzed	N/A	> 10 ⁶	1,500	133	N/A
Pt/Ti/NbO ₂ /Pt/Ti [R9]	Mott	Good, but not quantitatively analyzed	N/A	> 10 ⁹	800	392	0.1
Pt/VO ₂ /Pt [R10]	Mott	Good, but not quantitatively analyzed	7%	> 26.6×10 ⁶	> 60	11	0.01
TE/VO ₂ /BE [R11]	Mott	N/A	N/A	10 ⁹	200	11.9	0.1

Table R1. Comparisons with various memristor-based artificial neurons. To demonstrate the effectiveness of the gradual TiO_x memristor-based artificial neuron for the reliable and energy-efficient artificial neuron, several characteristics are compared to the existing artificial neuron devices. The gradual TiO_x memristor-based neuron shows advantages in terms of uniformity, low spike peak current, and high off-state resistance, and therefore, it is proper for the artificial neuron device with high reliability and low power consumption.

(6) Comparisons to the existing devices as a reservoir

We also provide comparisons to other devices as a reservoir in the Table R2. In Table R2, characteristics of several memristors that have been used for RC are compared to the gradual TiO_x memristor. As shown in the Table R2, the uniformity of the reservoir devices has been rarely studied, even though the cycle-to-cycle and device-to-device uniformity are essential for training and inference of the memristive RC system. The developed gradual TiO_x memristor has superior uniformity, high on/off ratio, high endurance, and fast speed, which are necessary properties for the memristor-based RC.

Device structure	Array size	On/off ratio	Cycle-to-cycle variation	Device-to-device variation	Pulse amplitude and width	Endurance	Rectification ratio
Pt/gradual oxygen concentration TiO_x/Ti	20×20 Crossbar array	> 2,000	1.39%	3.87%	Write : 4.5 V, 10 μs Read : 1.5 V, 100 μs	> 5×10^6	10^4
Au/Pd/ $\text{W}_x\text{O}_y/\text{W}$ [R12]	32×32 Crossbar array	N/A	N/A	N/A	Write : 1.5 V, 1 ms Read : 0.5 V, 500 μs	N/A	N/A
Au/Pd/ $\text{W}_x\text{O}_y/\text{W}$ [R13]	32×32 Crossbar array	N/A	N/A	N/A	Write : 3 V, 10 μs Read : 0.6 V, 200 μs	N/A	N/A
Pt/ $\text{SiO}_x\text{N}_y/\text{Ag}/\text{Pt}$ [R14]	Stand-alone	> 10^4	N/A	N/A	Write : 1.25 V, 100 μs Read : 0.1 V, 300 μs	> 10^6	N/A
Pt/ $\text{TaO}_y/\text{TiO}_x/\text{Ti}$ [R15]	Stand-alone	N/A	Good, but not quantitatively analyzed	N/A	Write : $-2 \sim 2$ V, 120 μs	N/A	10^3
Au/Cr/SnS flake/Au/Cr [R16]	1×10 array	< 2	N/A	N/A	Write : 4.5 V, 20 ms Read : 1 V, 25 ms	N/A	N/A

Table R2. Comparisons with other various memristors for reservoir computing. To demonstrate the effectiveness of the gradual TiO_x memristor as a RC device, several characteristics of the device are compared to the various memristors used for RC. The gradual TiO_x memristor satisfies the superior uniformity, high on/off ratio, fast speed, high endurance, and high rectifying ratio for sneak path problems simultaneously.

To clarify the improvements in terms of the artificial neuron and the RC system, we appended Table R1, Table R2, and corresponding sentences to Supplementary Table 1 and Supplementary Table 2.

In addition, the following sentences are appended in the conclusion section:

Page 22, line 474: “The newly developed memristor is utilized for memristor-based LIF neuron, and it shows several superior characteristics such as high uniformity, tunability, and low operation current (see Supplementary Table 1).”

Page 22, line 491: “The comparisons with the memristors in the previous works for the reservoir computing are shown in Supplementary Table 2.”

(7) Designed circuit for gradual TiO_x memristor-based RC

To show the feasibility of the memristor-based artificial neuron and RC, we propose a complete circuitry for the memristor-based artificial neuron and RC. The circuitry for the artificial neuron is already shown in the Figure R4 and R5. The complete circuitry of the proposed RC system in our study is in Figure R15.

The memristive RC system is composed of a highly reliable dynamic memristor array and peripheral circuitry. Each sequence element (character) is assigned to a different column of the cross-bar array. Each row of the cross-bar array is a single sub-reservoir, and memristors in different rows have different W/I ratios (duty cycles). The sequence data is transformed to the pulse train of desired duty cycles for each row by the pulse generator. The feeding voltage between set voltage and read voltage is chosen by the voltage selector. The switches determine the selected column for a character. The selected column is grounded, while others are floating. Each readout circuit, composed of an integrator and an analog-digital converter (ADC), is connected to corresponding column. The readout circuit measures the current of the memristors in a single row at the same time. The current level of each row will be different because of the different duty cycles. The long-term reservoir (the biggest W/I ratio) has the largest current level, while the short-term reservoir (the smallest W/I ratio) has the smallest current level. To compensate the difference of the current magnitude, two methods can be utilized; (1) The range of the output of an integrator can be controlled by different capacitances and a multiplexer (MUX), or (2) the longer read pulse can be used for the short-term reservoir to increase the charged data of the capacitor in the readout circuit. The outputs from each memristor device are mapped to each node of the readout function, and the readout function trains its weights with backpropagation method based on the **discrepancy** between the readout function output and the target data.

Figure R15. A circuit schematic for the memristive RC composed of the highly reliable dynamic memristor array and the peripheral circuit.

To inform the information about the designed circuitry for the RC system, we added sentences as following:

Page 19, line 412: “To simulate the memristive RC based on the measured data from memristor arrays, we design a whole system for memristive RC on a circuit level (see Supplementary Fig. 19).”

Figure R15 is added in Supplementary information as Figure S19.

(8) Improved abstract, introduction, and conclusion to include the results of the artificial neuron

Because the memristor for the reservoir computing also requires short-term memory effect and leaky-integrate properties of a neuron, we have expanded our work from RC devices to artificial neuron devices. The proposed device can be widely used from the bio-plausible neural networks to the reservoir computing. To reflect the changed focus correctly, we have revised the abstract, introduction, and conclusion sections.

We have revised the abstract as following:

Page 2, line 16: “Neuromorphic computing, a computing paradigm inspired by the human brain, enables energy-efficient and fast artificial neural networks. To process information, neuromorphic computing directly mimics the operation of a biological neuron, which has synaptic plasticity and firing properties. To effectively imitate biological neurons with electrical devices, memristor-based artificial neurons attract great attention because of their simple structure, energy efficiency, and excellent scalability. However, memristor’s non-reliability issues have been one of the main obstacles for the development of memristor-based artificial neuron and neuromorphic computing. Here, we show a memristor 1R cross-bar array without transistor devices for individual memristor access with low variation, 100% yield, large dynamic range, and fast speed for artificial neuron and neuromorphic computing. Based on the developed novel memristor, we experimentally demonstrate a memristor-based neuron with leaky-integrate and fire (LIF) property with excellent reliability. Furthermore, by using this newly developed memristor neuron’s leaky-integrate property, we develop a memristive reservoir computing (RC) system, a type of neuromorphic computing, for efficient processing of sequential/temporal data. Our new memristive RC system successfully trains and generates bio-medical sequential data (antimicrobial peptides) while using a small number of training parameters. Our results open up the possibility of a memristor-based artificial neuron and neuromorphic computing system, which is essential for energy-efficient edge computing devices.”

We have modified the introduction as following:

Page 3, line 34: “Artificial neural networks (ANNs) have shown their effectiveness in various fields such as autonomous cars, finding new drugs, designing circuits, or predicting protein structures¹⁷⁻²⁰. Powered by today’s elaborate complementary metal-oxide-semiconductor (CMOS) based computing processors, ANNs accomplish highly complex tasks. However,

ANNs running in a conventional von Neumann architecture computer (e.g. graphic processing units (GPU)) cannot reach the efficiency of a biological neural network due to the bottleneck of big data transfer⁸. To substitute the conventional ANNs, neuromorphic computing hardware, which imitates the operation of the brain and biological neural network, has been extensively studied. The conventional CMOS-based neuromorphic hardware uses complex CMOS circuitry to imitate the neuronal activity²¹. However, CMOS-based complex circuitry limits scalability and increases energy consumption. Thus, it is difficult to reach the efficiency of a biological neural network using CMOS-based neuromorphic hardware¹⁰.

Memristors, instead of the CMOS-based circuit, have been widely studied as a new candidate for neuromorphic hardware. Memristors are emerging memory devices that store information in the form of resistance by changing the internal distribution of oxygen anions or metal cations^{22,23}. They are a promising candidate for an artificial neuron device for neuromorphic hardware because of their great scalability, high energy efficiency, fast speed, small footprint, and simple fabrication process. Previous studies have developed several memristive artificial neurons and proved that memristors are proper devices for efficient artificial neurons. Short-term (volatile) memristors are used to represent two different biological neuron properties; a short-term potentiation and depression property, called paired-pulse facilitation (PPF) and paired-pulse depression (PPD), and leaky-integrate and firing (LIF) property of a biological neuron. Instead of being used for a synapse, short-term memristors are widely used for various neuromorphic computing systems such as reservoir computing (RC) and spiking neural network (SNN)^{2,3,13,14}.

Despite these advantages, the current memristor-based artificial neuron has several limitations. Various studies have proven the feasibility of memristor-based neuron, by unambiguously showing neuronal operations such as LIF and short-term plasticity. Among them, diffusive memristors ($\text{SiO}_x\text{N}_y:\text{Ag}$) or Mott memristors (NbO_2 or VO_2) have been extensively studied as artificial neuron device^{3,6,10}. The diffusive memristor type neurons are fast and have high on/off ratio, and the Mott memristor type neurons have acceptable uniformity and fast speed. However, the diffusive memristor type neurons have uniformity issues, and Mott memristors require a large operation current ($\sim\text{mA}$). In general, memristors usually suffer from large variation problems and unreliable switching²²⁻²⁴. These reliability problems degrade the memristor-based neuron performance and make it difficult to build large-scale memristor-based neuromorphic computing hardware. Even though reliable memristors may exist, it is difficult to operate a memristor-based cross-bar array without other components, such as selectors and transistors, because undesired current paths, which are called sneak paths, hinder the reliable read and set/reset operations in a memristor cross-bar array²². In addition, the low on/off ratio increases the complexity of the peripheral circuit that is needed to distinguish the small conductance change of the memristor, and high current requirement ($\sim\text{mA}$) increases the energy consumption. To overcome these bottlenecks, development of a new memristor that satisfies all of the requirements is warranted.

Here, we propose a transistor-free 1R structure memristor that consists of metal oxide with gradual oxygen concentrations that are fabricated in low-temperature environments for memristor-based artificial neuron and memristive RC system construction. We demonstrate that this memristor performs with high yield in array form ($\sim 100\%$), obtains self-rectifying behavior, has high temporal/spatial uniformity (1.39% and 3.87%, respectively), high

endurance without degradation ($> 10^6$), high speed (10 μ s), and a consistent time constant (~ 40 ms) in the array. Based on these ideal properties, we show that the developed **memristor-based artificial neuron possesses paired-pulse facilitation (PPF), paired-pulse depression (PPD), and leaky-integrate and fire (LIF) characteristics, which are the key characteristics of a biological neuron. The developed neuron has high spatio-temporal uniformity, which is one of the essential features for building a reliable memristor-based neuromorphic processor. In addition to the artificial neuron, we build a memristive RC system by using the memristors as leaky-integrate neurons.** The developed memristive RC can deal with sequential data, which can provide further complications compared to temporal data processing. With the developed memristive RC, anti-microbial peptides (AMPs), the anti-bacterial elements in the innate immune system, are utilized. The memristive RC, trained by AMP data, successfully learns the complex amino-acid grammar of AMPs and generates new AMPs.”

We have changed the conclusion as following:

Page 21, line 468: “In summary, we developed a novel 20 \times 20 high-density memristor array by utilizing gradual oxygen concentration metal oxide, which possesses a high on/off ratio, excellent temporal/spatial uniformity, self-rectification, forming-free property, compliance current-free property, and high yield for a memristor-based neuromorphic system. Unlike the other existing memristors operating with conductive filaments, our gradual TiO_x memristor switches its resistive state through oxygen migration without strong filament formation and changes the effective insulator thickness. It showed dynamic analog operations, such as potentiation (set), depression (reset), and self-decaying. The newly developed memristor **is utilized for memristor-based LIF neuron, and shows several superior characteristics such as high uniformity, tunability, and low operating current (see Supplementary Table 1).** It is demonstrated that the gradual TiO_x memristor-based artificial neuron has similar features to the biological LIF neuron by modifying the presynaptic spike or the capacitance of the artificial neuron. Based on the outstanding characteristics of the gradual TiO_x memristor-based neuron, our results open up a path to developing a reliable and energy-efficient neuromorphic processor with biological learning rules.

Furthermore, the gradual TiO_x memristor cross-bar array as a short-term, leaky-integrator is used to create a hardware-based reservoir computing system. The newly developed memristive RC system learns the AMP sequences and generates new AMP candidates. **To improve the performance of the memristive RC system, the multiple sub-reservoirs with different pulse width/interval ratios are utilized.** The **multiple sub-reservoir** effectively expands the reservoirs’ dimensionality, where short-term reservoirs memorize recent information with short decaying time constant while long-term reservoirs memorize information for a long period with long decaying time constant. **Multi-reservoir memristive RC with the gradual TiO_x memristor having excellent uniformity could be used for mass-production because the trained readout function will have weights that are matched to the reservoir in every single chip. Because of the reduced training cost by substituting the recurrent layer of the RNN into the reservoir, the memristive RC system can be suitable for online learning with high accuracy. The comparisons with the memristors in the previous works for the reservoir computing are shown in Supplementary Table 2. In addition, several superior properties, including self-rectification, forming-free, low temperature fabrication, and high on/off ratio make the gradual TiO_x memristor suitable for memristive RC system.**

The proposed highly reliable memristor can be broadly used in neuromorphic processors that handle bio-plausible neural networks such as spiking neural networks or reservoir computing. Further improvements such as peripheral control circuits, non-volatile memristor arrays as synapses, and the developed reliable artificial neurons for building SNN or RC system can be used for several practical applications including image classifications, speech recognitions, or real-time diagnosis with effectively reduced energy consumption and device size.”

Comment #5:

RC models have less parameters than LSTMs and RNNs. This statement holds true for LSTMs which have complex gating models, but they have an equivalent number of parameters as RNNs. The only other benefit for RC is the reduced training cost but from the discussion in the paper it does not focus on online learning. Rather they discuss a fixed, pre-trained readout which can be deployed to multiple crossbars due to the uniform nature of their fabrication.

Response: Thank you for the constructive comment. As the reviewer mentioned, the benefit of the RC compared to the RNN is the reduced training cost. We agree that it is essential to discuss the advantages of RC in training as well as inference.

To address the point, first, we have corrected the sentence stating RC has fewer parameters than RNN.

Page 15, line 304: “RC is a type of neuromorphic computing that processes sequential/temporal data. Conventional recurrent neural network (RNN) and a long short-term memory (LSTM), which are widely used neural networks for sequential/temporal data, have a complex and repeated recurrent layer, and the recurrent layer makes the training cost increase. On the contrary, in the RC, the “reservoir” transforms sequential/temporal data into a high dimensional basis by using a short-term memory effect of the reservoir without complex calculations. The input data, processed by the reservoir, is fed to the readout function. The readout function is usually a simple fully connected network and trains its weights by using a conventional machine learning algorithm.”

In addition, we have appended the advantages of the developed RC system for online training with the reliable memristor-based hardware. Cycle-to-cycle variation (or temporal variation) induces errors in the reservoir output for every input, and this variation may limit the online training. However, because the newly developed memristor has superior uniformity, stable online training without significant errors in the reservoir can be achievable. We have added the sentence as following:

Page 22, line 487: “Multi-reservoir memristive RC with the gradual TiO_x memristor having excellent uniformity could be used for mass-production because the trained readout function will have weights that are matched to reservoir in every single chip. Because of the reduced training cost by substituting the recurrent layer of the RNN into the reservoir, the memristive RC system can be suitable for online learning with high accuracy”

Comment #6:

Beyond that, at a high level the claim this as an RC computing model requires serious validation. From all the tests and architectures described there is no recurrent network. Rather, from the way it is presented, a set of memristors that act as leaky integrators are implemented and readout the final state of each individual memristor for classification. This is not a true RC model, and it is not clear how they treat the inputs to the network in terms of weights. In writing they discuss how the memristors see a combination of inputs, but in the alphabet example it is described as a 1-to-1 mapping where each memristor only sees a single input. A similar setup is described for the AMP sequences.

Response:

Thank you for your valuable comment. As the reviewer pointed out, the developed RC system does not have recurrence among different characters, but the recurrence only exists for each character. This is similar to the RNN which has a weight matrix that is composed of non-zero diagonal elements while other elements are zero; for example, if the sequence is a multi-dimensional vector, where the length of the vector is the length of the sequence and the dimension is the number of unique characters, there is no recurrence among each dimension. Therefore, we used a simple MLP with an activation function, because there is no non-linear connection between different characters (dimensions). In this point of view, the developed RC system is not an ideal RC, as the reviewer commented.

Even though the RC system in this work is not an ideal RC, similar setups have been demonstrated for some applications. In a pioneering study, a memristor-based RC system has classified the number pronounced by human voice (Moon, J. et al. [*Nat. Electron.*, 2019]) and this work utilizes a similar setup (1-to-1 mapping of frequency range and memristor). In the work by Moon, the voice data is transformed into time-frequency data having a length in 50 (time step) and 50 channels (dimensions), as shown in the Figure R16. Each channel represents a specific frequency region and is assigned to one of the memristors. By using this method, the study obtained 99.2% classification accuracy and the result shows that matching a single variable to a single memristor can be used for RC handling multivariable data. However, as the reviewer mentioned, there is no recurrence among different channels in this type of RC system. Therefore, this approach has a limitation if the recurrence among different channels, dimensions, or characters is significant for the used data. In our work, even though this approach is not an ideal RC, it is still effective for some sequence data, which is barely studied with memristive RC, unlike single-variable time-series data. It shows great performance for the given sequence data (antimicrobial peptides). We modified the manuscript that might cause misunderstandings about the non-ideal RC system.

Figure R16. Classification of spoken digits with memristive RC in the work by Moon, J. et al. [Nat. Electron., 2019]. a and b, The spoken digit data has 50 channels (frequency domain) and 50 lengths (time step) and it is converted into the digitized spike trains. **c**, The temporal responses of the memristors. The memristors are assigned for each channel. **d**, The classification results from the memristive RC.

To clarify that the unique character in the sequence is considered as a channel or a dimension of a multivariable vector, we added the Figure R17 as a Figure 5b and corresponding sentences to show that the sequence data used for the RC is transformed into a multi-dimensional vector where each unique character represents each dimension of the vector:

Page 19, line 417: “First, the input sequence is transformed into a 20-dimensional (or 20-channel) vector having the same time step to the original sequence data. Each memristor in a reservoir processes the spike train from the one channel (one amino-acid) among the 20 channels in the transformed input data (see Figure 5b).”

Figure R17. An illustration of the transformation of an input sequence into a 20-dimensional vector (or 20-channels) with 24-time steps, where 20 is the number of the unique amino-acids in the input sequence with the added end-cursor “@”, and 24 is the length of the sequence after adding the end-cursor.

Furthermore, as the reviewer commented, there is little information about how input data is treated at the readout function, or the neural network level. After the input data is processed in the reservoir, the reservoir output is the input of the readout function, which is a simple multi-level perceptron neural network of the reservoir computing system. Because there are four reservoirs and each of them has 20 memristors, the reservoir output is a 1×80 vector. This vector is the input of the readout function, and the readout function trains its weights by using the next amino-acid element as a target (label). The readout function has a three-layered structure with 80, 500, and 20 neurons for each layer. The first layer is the reservoir output itself, thus, the layer size is 1×80 . The first and second layers use a ReLU as an activation function while the last layer uses a soft-max as an activation function. Even though there is no connection among each variable in a reservoir, the RC system well trains its weights, because each node of the reservoir output is non-linearly connected in the readout function. A categorical cross-entropy is used for the loss function, which is widely used for prediction or classification tasks. In summary, the RC system learns the amino-acid grammar of the AMP sequence, by predicting the next input from the training data.

To inform this information, we have added the Figure R15 in Supplementary Fig. 19, which shows how the memristive RC system receives input and processes the input with the reservoir and the readout function.

Comment #7:

By framing the device contributions in the light of new neuron circuits and generalizable models, there could be lot more merit to this paper, rather than presenting it as an RC memristor.

Response: Thank you very much for the constructive comment and for suggesting the use of our device as an artificial neuron. As the reviewer suggested, the proposed device can be used as an artificial neuron for several important future applications such as bio-mimicking robotics, spiking neural networks, and reservoir computing. The manuscript now is focused on the two results; the artificial neuron that mimics the biological LIF neuron model, and the memristor-based reservoir computing system that requires short-term neurons (or leaky-integrators) as a reservoir. To demonstrate that the gradual TiO_x memristor can be utilized as an artificial neuron and to show the advantages of the neuron, we added the Figure 3 in the main text of the manuscript, as shown in the Figure R18. The corresponding subsection “Reliable LIF neuron with gradual TiO_x memristor” is added in the main text, as we discussed in the **comment #4**.

Figure R18. Gradual TiO_x memristor-based LIF neuron.

Comment #8:

Another concern is that the authors discuss the high-dimensional basis used for classification, but if every memristor is only a filtered version reflecting the activity of a single input category, each device belonging to the same input should be the same. This is almost like concatenating the input multiple times, but not truly expanding to a high-dimensional space where the input would become linearly separable. In this setup, one cannot see the need to use more devices than the number of unique input characters.

Response: Thank you for this helpful comment. The memristors with different W/I ratios have different potentiation ratios and decaying time constants. As shown in the Figure R19, the potentiation ratio and the decaying speed are dependent on W/I ratio. Therefore, even though the memristors receive the same input, they do not have linear relationships; the history of each character differently affects the memristor state because of the different potentiation ratios and decaying speeds.

As shown in Figure R20 (Figure 3d-f in the original manuscript), the final outputs are different due to different W/I ratios, even though the inputs are identical. If the W/I ratio is high, it stores the history of the character strongly because of the slow decaying speed. In the given sequence “DDDCBDCDBADBACDCBA”, the character “D” is the most frequently appearing character, and “A” is the least frequently appearing character. Therefore, with high W/I ratio (85/15), the largest output is “D”. If the W/I ratio is low, it represents the recent information. As a result, the largest output is “A” with low W/I ratio (15/85).

This example shows that the memristors with different W/I ratios process the same input differently; for example, if a vector $m_i(A, B, C, D)$ is outputs of memristor for “A”, “B”, “C”, and “D” with W/I ratio “i”, then, $m_i(A, B, C, D) = c \times m_j(A, B, C, D)$ is not satisfied for different i and j (c is a positive constant number). Therefore, if the multiple W/I ratios are utilized for the reservoir, different outputs will be obtained from the memristors with different W/I ratios for the same input, which can improve the training of the readout function. This is a clear improvement compared to the prior studies (Moon, J. et al. [*Nat. Electron.*, 2019], and Chao Du et al. [*Nat. Commun.*, 2017]) which utilize the device-to-device variation. Device-to-device variation makes the learning easier, but the readout function must be re-trained when the different memristors are used. Therefore, it is difficult to use the pre-trained input for memristive reservoir computing. However, in our work, varying W/I ratio with reliable memristors can utilize the same pre-trained readout function for memristive reservoir computing.

Figure R19. The normalized responses of the gradual TiO_x memristor during the potentiation and decaying with various W/I ratios. The decaying time constant increases as the W/I ratio increases. The minimum and the maximum time constants are ~ 15 ms (W/I ratio $< 10/90$) and ~ 40 ms (W/I ratio $> 40/60$), respectively. The W/I ratio for saturation can be changed by the pulse amplitude and the unit pulse width.

Figure R20. Memristive reservoir output through the whole input sequence in a short-term, middle-term, and long-term reservoir. The final results of each reservoir output after all sequence input enters the reservoirs show the different processing abilities of each reservoir.

However, as the reviewer pointed out, the expansion of the dimension is limited because of no connection between each character, it is not proper to claim a high dimensional basis. Therefore, we removed the term “high dimensional basis” in the revised manuscript. In addition, to inform how the memristors with different W/I ratios can process the input differently, Figure R19 is included in the revised supplementary information as a Figure S17.

We also added the following sentences in the revised manuscript:

Page 17, line 356: “Interestingly, the duty cycle (for write pulse only) controls the decaying speed and changes the response of the memristor for the same input (see Supplementary Fig.17 and Supplementary Fig.18). As shown in Supplementary Fig. 17, a high pulse width/interval

(W/I) ratio results in a slow decaying speed (long decaying time constant), while a low pulse W/I ratio results in a fast decaying speed (short decaying time constant). The decaying time constant can be controlled from 15 ms to 40 ms by varying the W/I ratio, and the different decaying speed enables the memristors to process the same input differently.”

Page 17, line 366: “In the reservoir with a high W/I ratio (W/I ratio = 85/15), the frequent inputs dominantly affect the current state of the reservoir because of the long decaying time constant.”

Page 17, line 369: “Furthermore, in the short-term reservoir (W/I ratio = 15/85), the recent inputs strongly affect the current reservoir state and preceding inputs diminish due to the short decaying time constant. The elements at the end of the sequence, thus, dominate the outcome and the memristor for the last input “A” has the largest output, as shown in Fig. 4d.”

Page 17, line 375: “Therefore, if the multiple W/I ratios are utilized for the reservoir, different outputs will be obtained from the memristors with different W/I ratios for the same input, which can improve the training of the readout function.”

Comment #9:

Beyond this, how would the authors intend to extend the proposed approach to a single variable, continuous time-series data, or multivariable continuous data.

Response: Thank you for this valuable comment. Single variable continuous time-series data have been used in several previous studies (Moon, J. et al. [*Nat. Electron.*, 2019], and Chao Du et al. [*Nat. Commun.*, 2017], Zhong, Y. et al. [*Nat. Commun.*, 2021]). For single-variable data, the input is transformed into a quantized pulse train with different amplitudes. It is possible to process a single variable continuous time-series data with the gradual TiO_x memristor, only if the input data is transformed into a pulse train with various amplitudes. The gradual TiO_x memristor-based reservoir computing system has an advantage because the prior studies utilize the device-to-device variation for learning while we utilized the W/I ratio method, which enables the use of the pre-trained readout function.

Unlike the single-variable continuous time-series data, it has not been reported that multivariable data have been processed by nanodevice-based RC. As the reviewer pointed out in the comments #6 and #8, the 1-to-1 matching method is not enough to transform the input into a high-dimensional basis. However, some previous studies have processed multi-channel data by dividing image or voice data into several channels (Midya, R. et al. [*Adv. Intell. Syst.*, 2019], Moon, J. et al. [*Nat. Electron.*, 2019]), and they have proven that assigning a specific channel or dimension to memristors can effectively classify the inputs. A sequence can be considered as a type of multivariable data, where each unique character is a dimension and the sequence length is a data length, as shown in Figure R21. Therefore, the memristive RC system in our work can process the multivariable data, if the proper peripheral circuitry is utilized. For

example, a DAC is needed for the multivariable data, because a pulse train should have various pulse amplitudes to represent a quantized value of the data.

Figure R21. An illustration for comparison between sequence and multivariable data. Sequence data can be considered as a type of multivariable data, where each element (alphabet in this case) represents each dimension.

Comment #10:

That being said, in terms of clarity I think the paper is poorly written in that it does not flow. It was very hard to follow the discussion and figures, and there seemed to be very little discussion around several key topics.

First, the classification method relied on the decay of memristor states. If the decay is inherent to the memristor, it is unclear how it can be controlled to allow the memristor to hold a specific resistant state (the paper states that "decay from LRS to HRS could occur"). If it is not possible, there are limitations in how long information can be stored (likely dependent on the W/I ratio) but this is not thoroughly explored, and this device cannot be used as a synaptic weight which is common in memristor based neuromorphic circuits making it serve a very different purpose. If the decay can be controlled, what is the overhead associated with it?

Response: Thank you for this constructive comment. We agree that there were a lack of information about the device retention, and thus this might interrupt readers from following and understanding the work. As the reviewer mentioned, this device does not have long-term retention, and it cannot be used as synaptic weight. However, this short-term memory effect is useful for reservoir computing and artificial neuron, and the improvement of reliability is one of the important factors for a reservoir computing and an artificial neuron.

To clarify these points, we have removed ambiguous expressions in the original manuscript, and we added sentences to notice the device is volatile and it cannot serve as weight:

Page 3, line 51: “Short-term (volatile) memristors are used to represent two different biological neuron properties; a short-term potentiation and depression property, called paired-pulse facilitation (PPF) and paired-pulse depression (PPD), and leaky-integrate and firing (LIF) property of a biological neuron. Instead of being used for a synapse, short-term memristors are widely used for various neuromorphic computing systems such as reservoir computing (RC) and spiking neural network (SNN).”

The overhead associated with the decay control is that the unit pulse length must be longer than the minimum pulse length to make different W/I ratios (different widths and intervals) for RC. For example, the gradual TiO_x memristor can operate with 10 μs pulse, but the unit pulse length for various W/I ratios should be longer than 10 μs to make various W/I ratios such as 10 μs width and 90 μs interval or 20 μs width and 80 μs interval. In addition, the decaying speed can be controllable from 15 ms to 40 ms by using different W/I ratios, as shown in Figure R22. The same trend is observed for binary pulse train input, as shown in Figure R23. Figure R23 is normalized memristor responses from binary input pulse train with various W/I ratios (20/80~95/5). The unit pulse length is 100 μs . As shown in the figure, for the low W/I case, the potentiation is more abrupt and the decaying is fast, while the potentiation is gradual and the decaying is slow for the high W/I ratio case.

To inform this information, we added Figure R23 as a Supplementary Figure 18, and modified the sentence in the original manuscript (“Interestingly, the duty cycle (for write pulse only) controls its processing ability in terms of the number of processed elements.”) into the below sentences:

Page 17, line 356: “Interestingly, the duty cycle (for write pulse only) controls the decaying speed and changes the response of the memristor for the same input (see Supplementary Figure 17 and 18). As shown in Supplementary Figure 17, a high pulse width/interval (W/I) ratio results in a slow decaying speed (long decaying time constant), while a low pulse W/I ratio results in a fast decaying speed (short decaying time constant). The decaying time constant can be controlled from 15 ms to 40 ms by varying the W/I ratio, and the different decaying speed enables the memristors to process the same input differently.”

Page 17, line 375: “Therefore, if the multiple W/I ratios are utilized for the reservoir, different outputs will be obtained, which can improve the training of the readout function”,

Figure R22. The normalized responses of the gradual TiO_x memristor during the potentiation and decaying with various W/I ratios. The decaying time constant increases as the W/I ratio increases. The minimum and the maximum time constants are ~ 15 ms (W/I ratio $< 10/90$) and ~ 40 ms (W/I ratio $> 40/60$), respectively. The W/I ratio for saturation can be changed by the pulse amplitude and the unit pulse width.

Figure R23. The normalized memristor output current according to the binary pulse train with various W/I ratios. The unit pulse length is $100 \mu\text{s}$, and the W/I ratio increases from 20/80 to 95/5 with a $5 \mu\text{s}$ interval. When the W/I ratio is low, the potentiation and decaying are abrupt. On the contrary, the potentiation and decaying are gradual in the high W/I ratio cases. The results show that memristors with different W/I ratios process the same sequential input differently, because of the different potentiation ratio and the decaying speed.

Comment #11:

The other part that is difficult to understand is what the multi-modal RC setup is. It seems like it just refers to a system of crossbars with different W/I ratios which is confusing when used with the term multi-modal in ML.

Response: Thank you for this helpful comment. The word multi-modal RC is used because our RC system has multiple reservoirs which process input differently. However, as the reviewer pointed out, this term could mislead readers because the term multi-modal has a different meaning for the machine-learning field.

To address this problem, we change the word “multi-modal RC” into “multi-reservoir RC” and “RC with multiple sub-reservoirs” to clarify that this method uses multiple sub-reservoirs.

Comment #12:

But then questions arise such as: how scalable is the approach with varying W/I intervals, how large are the devices, what are the limits in terms of ratios and their respective memory lengths.

Response: Thank you for this constructive comment. This comment is are very important information which needs to be discussed in the manuscript. In this work, memristors in a short-term, middle-term, and long-term reservoir process input differently due to the different W/I ratios. However, this method sacrifices additional time, because various pulse widths and intervals should be included in a single period. As shown in Figure R24, when the width increases 5 μs (or the interval decreases 5 μs), the potentiation ratios and decaying speeds are clearly different. Therefore, if there is at least 5 μs width and interval difference for each reservoir, the reservoirs process the input differently to help the training of the readout function.

Figure R24. The normalized memristor output current according to the binary pulse train with various W/I ratios. The unit pulse length is 100 μs , and the W/I ratio increases from 20/80 to 95/5 with a 5 μs interval. When the W/I ratio is low, the potentiation and decaying are abrupt. On the contrary, the potentiation and decaying are gradual in the high W/I ratio cases. The results show that memristors with different W/I ratios process the same sequential input differently, because of the different potentiation ratios and the decaying speed.

To clarify this, we have added Figure R24 into the revised Supplementary information as a Figure S18, and added sentences to discuss the limitation of the W/I ratio method as following:

Page 17, line 377: “As shown in Supplementary Fig. 18, if the change of W/I ratio is larger than a minimum required time, the change in potentiation ratio and decaying time constant can be clearly observed.”

As the reviewer commented, the device size must be informed. For the memristor cross-bar array used for the reservoir computing, the device size is $5 \times 5 \mu\text{m}^2$. For the LIF neuron demonstration, $10 \times 10 \mu\text{m}^2$ size stand-alone devices are used. Due to the limitation of the photo lithography tool, the device was measured down to $5 \times 5 \mu\text{m}^2$.

To inform the device size used for each result, we add an explanation about the device size for each experiment as followings:

Page 16, line 335: “All experiments are conducted in a 20×20 gradual TiO_x memristor cross-bar array, where a single cell size is $5 \times 5 \mu\text{m}^2$.”

Page 11, line 221: “Different pulse intervals (200 μs and 1500 μs) are applied to the $10 \times 10 \mu\text{m}^2$ gradual TiO_x memristor, while the input voltage amplitude and width (3 V, 100 μs) are kept constant.”

Finally, there is a limitation about the memory length of memristor-based RC due to the volatile nature of the device. For generation tasks or prediction tasks, the readout function needs to operate for every time step, because the next time step’s input needs to be printed out through

the readout function. The speed of this process (readout function operation) can be varied by the system environment. If the period for the readout function is longer than the memory length of the volatile memristor in the reservoir, the reservoir loses a lot of information. Therefore, if the readout function operation is slow and the reservoir has fast decaying time constant, the accuracy will significantly deteriorate. Therefore, the pulse width, interval width, and the W/I ratio must be optimized. To solve this problem, previous study (Moon, J. et al. [*Nat. Electron.*, 2019]) has used a long programming pulse with high amplitude to prevent the information in the device from being diminished. We are investigating another way for the future work to solve the problem by using a “hold” pulse that is a small set pulse and potentiates the memristor little bit to prevent the memristor from losing its entire information.

Comment #13:

Grammatical edits:

Line 27 - while using a small number of training

Line 60 - The mentioned bottlenecks can be reiterated and mapped to their solutions in the existing work in the later stages of the paper

Line 170 - anions across the whole TiOx bulk

Fig. 2 - (b) and (c) label spelling (electrode)

Response: Thank you very much for correcting the spelling in the original manuscript and suggestion to emphasize the improvements. Based on the reviewer’s correction, we revised the errors in the original manuscript as follows:

(1) Line 27 – while using a small number of training

Page 2, line 28: “Our new memristive RC system successfully trains and generates bio-medical sequential data (antimicrobial peptides) **while using a small number of training** parameters”

(2) Line 60 – The mentioned bottlenecks can be reiterated and mapped to their solutions in the existing work in the later stages of the paper

Page 6, line 101: “This high rectifying ratio (current at V_{read} / current at $-V_{\text{read}}$) prevents the selector-less cross-bar array from the sneak path current problems (see Supplementary Fig. 1), **and the high on/off ratio ($>10^3$) reduces the peripheral circuit burden to discriminate the conductance change, which are the major bottlenecks in the conventional memristor cross-bar array.**”

Page 6, line 113: “**These results show the high uniformity of the developed device, which has been a significant bottleneck for various memristor devices.**”

(3) Line 170 - anions across the whole TiOx bulk

Page 9, line 192: “All of these results demonstrate that resistance switching in gradual TiO_x is induced by the movement of oxygen anions across the whole TiO_x bulk, unlike other filamentary type memristors.”

(4) Fig. 2 - (b) and (c) label spelling (electrode)

We corrected the typo, which should be “Electrode” in the Figure 2b and 2c in the revised manuscript.

Reviewer #2

The work by Park et al. presents a TiO_x -based memristor system and proposed an RC system based on such memristors. There are clearly interesting results, including the forming-free and uniform switching characteristics of the TiO_x memristors. The materials study also seems solid, and the topic is timely. However, several concerns need to be addressed.

Response: We sincerely thank the reviewer for positive comments on the proposed memristor device. Based on the reviewer’s comments and suggestions, we have revised expressions that might be unclear and overreaching, and we have added more detailed explanations about the experimental setup. In addition, to further improve the effectiveness of the developed device, we experimentally demonstrated the artificial neuron based on the device. Our detailed responses to your comments are provided below.

Comment #1:

RC systems are suitable for processing temporal data. It’s thus puzzling why the authors picked the AMP task, which has no temporal components. In fact, the readout layer seems quite elaborate (multi-layer perception with nonlinear activations) already. Given the complex readout layer, instead of using an RC system, I would expect a conventional CNN network + the existing readout layer can probably achieve similar or better results.

Response: Thank you for your constructive comments. As the reviewer commented, RC generally used for temporal data, such as voice classification or stock price prediction. The reason why we chose the AMP task is that the processing of single-variable continuous time-series data by using memristor-based RC have been widely studied by previous works (e.g. Moon, J. et al. [*Nat. Electron.*, 2019], Zhong, Y. et al. [*Nat. Commun.*, 2021]), but there is little study for the sequential data, even though several important data exists as in the form of a sequence (e.g. text, protein sequence, DNA, etc.). Sequential data is also appropriate for the RC because, similar to the temporal data, a recent element or word in a sequence is strongly affected by the past element. Therefore, we have picked up the AMP task to show that the memristor-based RC can process sequential data.

In addition, as the reviewer pointed out, the conventional convolutional neural network (CNN) will achieve much better accuracy for some tasks such as classification or object detection. However, CNN requires large computation resources to calculate each convolution operation for each receptive field, and to store the feature map generated from each layer. Therefore, even though the CNN is a powerful neural network, it might not be proper for energy efficient AI or edge computing. On the other hand, RC can effectively process the input data without complex computations. The training cost can be reduced significantly and the network becomes simple without large hardware burden. Therefore, RC can be the one of the strong candidates for limited computational resources environment such as an edge computing system.

Because we utilize a non-linear activation function (ReLU) in the readout function which is a simple multi-layer perceptron, the computational complexity is higher than the traditional RC that uses a linear readout function. However, the reservoir with three-layer MLP with ReLU as an activation function is still much simpler than other networks such as CNN, RNN, and LSTM. The detailed explanations about why the MLP is used as a readout function are addressed in the reviewer's comment #3.

Comment #2:

The AMP task is achieved through simulation. The authors have enough memristors and have 100% yield, so why not implement it experimentally?

Response: Thank you for this constructive comment. We strongly agree that it would be more beneficial if we experimentally demonstrate the memristor-based RC hardware. Unfortunately, however, hardware implementation of the memristor RC requires much more optimization of several elements such as readout circuits, analog-to-digital converter (ADC), control unit, etc. As the reviewer commented, it is important to demonstrate the hardware system, we are currently working on the hardware implementation of the memristor-based RC as the following future work.

Comment #3:

The experimental setup was very unclear. There was no discussion in the main text, and only scant information in “Methods”. From Supplementary Information Figure S13, it seems the authors send 1 character to a corresponding memristor, instead of sending a sequence to 1 memristor. Why? Overall a careful justification for the experimental setup needs to be presented, and it should be included in the main text.

Response: Thank you very much for these valuable comments and suggestions. As the reviewer pointed out, there is little discussion and explanations about the experimental setup, and it might cause misleading to readers.

To clearly represent how the RC operates and why each memristor is assigned for each unique character, we have added additional explanations and figures about the experimental setup and the operation principles of the RC in the revised manuscript.

(1) Why each memristor receives 1 character instead of a sequence

For a sequence input, it is not possible to put the whole input into a single memristor because each element (or character in this case) cannot be encoded with amplitude. In other words, a single memristor can process a single-variable continuous data by applying the quantized version of the input to the memristor in form of the voltage pulse, but it cannot process data with multiple channels or characters. Therefore, when the input is a sequence or multivariable data that consist of multiple dimensions, each character or dimension needs to be sent to the assigned memristors. Several prior studies have used similar methods (e.g. Midya, R. et al. [Adv. Intell. Syst., 2019], and Moon, J. et al. [Nat. Electron., 2019]) for processing image data or voice data. For example, voice data are transformed into a frequency domain and each frequency region is assigned to a memristor. This method is effective when the data have multiple channels or dimensions, and a sequence can be considered as multivariable data with multiple dimensions (unique characters), as shown in Figure R25.

Figure R25. An illustration for comparison between sequence and multivariable data. Sequence data can be considered as a type of multivariable data, where each element (alphabet in this case) represents each dimension.

Similar to the multivariable data, in our experiment, each memristor is assigned to each unique character and the response of the memristor contains information about the history of each

character. The responses of the memristors are used to train the readout function. In the sub-reservoir, the memristors can process the same input differently because decaying speed and potentiation ratio can be controlled by various W/I ratios, as shown in Figure R26. By using the multiple memristors with different W/I ratios for a single input, the sequence input can be effectively processed by the memristor-based reservoir.

Figure R26. The normalized responses of the gradual TiO_x memristor during the potentiation and decaying with various W/I ratios. The decaying time constant increases as the W/I ratio increases. The minimum and the maximum time constants are ~ 15 ms (W/I ratio $< 10/90$) and ~ 40 ms (W/I ratio $> 40/60$), respectively. The W/I ratio for saturation can be changed by the pulse amplitude and the unit pulse width.

To inform this information, we added sentences to elaborately explain the experimental setup, the reason why the memristor is assigned to a single character, and how the memristive RC operates in the revised manuscript.

Page 17, line 356: “Interestingly, the duty cycle (for write pulse only) controls the decaying speed and changes the response of the memristor for the same input (see Supplementary Fig. 17 and 18). As shown in Supplementary Fig. 17, a high pulse width/interval (W/I) ratio results in a slow decaying speed (long decaying time constant), while a low pulse W/I ratio results in a fast decaying speed (short decaying time constant). The decaying time constant can be controlled from 15 ms to 40 ms by varying the W/I ratio, and the different decaying speed enables the memristors to process the same input differently.”

The corresponding Figure R26 is added in a Supplementary Figure 17.

(2) The experimental setup for the memristive RC

To provide a clear representation of the experimental setup of memristive RC, we have added a designed circuitry of the memristive RC and the illustration for RC operation, as shown in Figure R27 and R28.

The memristive RC system is composed of the highly reliable dynamic memristor array and the peripheral circuit, as shown in Figure R27. Each sequence element (character) is assigned to a different column of the cross-bar array. Each row of the cross-bar array is a single sub-reservoir, and memristors in different rows have different W/I ratios (duty cycle). The sequence data is transformed to the pulse train of desired duty cycles for each row by the pulse generator. The feeding voltage between set voltage and read voltage is chosen by the voltage selector. The switches determine the selected column for a character. The selected column is grounded, while the others are floating. Each readout circuit, composed of an integrator and an analog-digital converter (ADC), is connected to the corresponding column. The readout circuit measures the current of the memristors in a single row at the same. The current level of each row will be different because of the different duty cycles. The long-term reservoir (the biggest W/I ratio) has the largest current level, while the short-term reservoir (the smallest W/I ratio) has the smallest current level. To compensate the difference of the current magnitude, two methods can be utilized; (1) The range of the output of an integrator can be controlled by different capacitances and a multiplexer (MUX), or (2) the longer read pulse can be used for the short-term reservoir to increase the charged data of the capacitor in the readout circuit. The outputs from each memristor device are mapped to each node of the readout function, and the readout function trains its weights with the backpropagation method based on the **discrepancy** between the readout function output and the target data.

Figure R27. A circuit schematic for the memristive RC composed of the highly reliable dynamic memristor array and the peripheral circuit.

Figure R28. An illustration of the operation of the developed memristive RC system.

The operation of RC is illustrated in Figure R28. In the figure, input character "G" enters the reservoir and the column for the "G" in the memristor cross-bar array is grounded to potentiate the column "G". The different pulse duty cycle (W/I ratio) for each row is applied for sub-reservoirs. During read process, the read pulses are applied to the row lines while all column lines are grounded. The conductances of the memristors in the cross-bar array are transferred to the input of the readout function. Based on the discrepancy between the readout function's

output and the next target “I”, followed by the input “G”, the readout function trains its weights. This protocol is repeated until the end of the input sequence.

To inform this information, we have added following sentences in the revised manuscript.

Page 19, line 417: “First, the input sequence is transformed into a 20-dimensional (or 20-channel) vector having the same time step to the original sequence data. Each memristor in a reservoir processes the spike train from the one channel (one amino-acid) among the 20 channels in the transformed input data (see Figure 5b). In our system design, the reservoir consists of four different sub-reservoirs with various W/I ratios to process the given input data differently (see Supplementary Fig. 19, 20, and Methods). To train the memristive RC to generate the new AMPs, the training data is preprocessed first. The target data to train the weights in the readout function is generated from the training data. For example, for the input sequence in Fig. 5b, the target data is made by removing the first amino-acid “G” and adding an end-cursor “@” at the end of the sequence, thus the target data becomes “IGKFLHSAKKFGKAFVGEIMNS@”. The end-cursor is used to inform the sequence end. The first amino-acid, which is “G”, enters the memristor cross-bar array. Every column line represents each amino-acid used for AMP, and each row line represents a single reservoir, as shown in Supplementary Fig. 19 and 20. When the input amino-acid is “G”, the column line for “G” is grounded and the other columns are floating. Then, the voltage pulses with various W/I ratios are transported from the row line and potentiate the memristors in column “G”. After the potentiation of memristors for “G”, a read stage is performed by applying the read voltage to the row lines while the column lines are grounded. The reservoir output has 1×80 size because each sub-reservoir makes a 1×20 output and there are four sub-reservoirs. The reservoir output is an input of the readout function, and the readout function trains the weights based on the discrepancy between the readout function output and the corresponding target, which is the next amino-acid “I” in the given example. By repeating this progress until the end-cursor enters the reservoir, the memristive RC can learn the amino-acid grammar for the AMPs.”

we have added corresponding figures Figure R27 and R28, as Supplementary Figure 19 and 20.

Comment #4:

The authors highlight throughout the manuscript the idea of “gradual TiO_x” and showed the two regions of amorphous oxygen-rich layer and polycrystalline oxygen-deficient layer. Since the TiO_x layer is created through a single anodization of the Ti BE, why is the bottom layer polycrystalline while the top layer amorphous?

Response: Thank you for this valuable comment. The Figure R29 and R 30 are the TEM image and the SIMS results, which are the Supplementary Fig. 8b and Fig. 2b and 2c in the original manuscript, respectively. From the Figure R29, poly-crystalline short-range lattice structures are observed at the TiO_x region near bottom electrode (BE), while no lattice structure is observed at the upper TiO_x region (an amorphous region). Because metallic Ti is rich in the

bottom layer of TiO_x as SIMS results in the Figure R30 demonstrate, the rich metallic Ti strongly affects to poly-crystalline phase and lattice structure of the bottom TiO_x layer. The oxygen-rich part of TiO_x layer shows amorphous phase due to low metallic Ti concentration. Therefore, the different distribution of metallic Ti determines the amorphous top layer and polycrystalline bottom layer.

Figure R29. TEM image of the gradual TiO_x memristor.

Figure R30. TOF-SIMS depth profile results of anodized TiO_x and sputtered TiO_2 .

To clarify this point, we have added an explanation of why the phases between the upper and bottom regions are different:

Page 9, line 186: “Moreover, in the anodized TiO_x layer, it is observed that the upper region of the TiO_x has a fully amorphous structure while the bottom region has a polycrystalline structure, because polycrystalline metallic Ti concentration is much higher in the bottom region of the TiO_x layer (see Supplementary Fig. 8 and 9).”

Comment #5:

Several claims are over-reaching and incorrect. Including “To the best of our knowledge, this is the first study to demonstrate a large 1R memristor crossbar array with 100% device yield.” (p6), and “For the first time, we trained and generated sequential data with memristive RC.” (p15).

Response: Thank you for correcting the claims. The reviewer’s comment is valuable and we agree that it is necessary to tone down the claims.

To reflect the reviewer’s suggestions, we removed the two over-reaching sentences in page 6 and page 15 from the revised manuscript.

Reviewer #3

This work combine two highly hot topics: memristors and machine learning. The main novelty of the work consist in application of memristors' array, while individual memristors possess average characteristics in comparison with other recently published based on TiO₂ (10.1063/1.5037835, 10.3389/fchem.2020.00724,10.1063/1.4940361) and other materials (10.1038/s41467-021-25455-0,10.1038/s41586-021-03748-0). Authors demonstrates important results and this study hs significant contribution to the field of application of memristors in machine learning. I recommend to publish this article in Nature Communications after minor revision. Several questions are listed below

Response: We sincerely appreciate the reviewer for positive comments on this work. Based on the reviewer’s comments and suggestions, we have conducted X-ray test, clarified the expressions and explanations. In addition, to further improve the effectiveness of the developed device, we experimentally demonstrated the artificial neuron based on the device. Our detailed responses to your technical comments are provided below.

Comment #1:

**1) how was obatin fig. 1f,g and fig. S3? Is this photo or schemes?
I recommend also provide SEM of crossbar array or AFM**

Response: Thank you for this helpful comment. The image in the Figure 1f, g, and Supplementary Figure 3 is an optical image of the fabricated cross-bar array. As the reviewer commented, it could be ambiguous that the cross-bar figures are whether photos or schemes. Therefore, the SEM image and the optical image of the cross-bar array are shown together in Figure R31, and added in the revised manuscript.

Figure R31. A scanning electron microscope (SEM) image (a) and an optical microscope image (b) of the fabricated cross-bar array. The size of a single cell in the array is $5 \times 5 \mu\text{m}^2$. The column lines are bottom electrodes (Ti) and the row lines are top electrodes (Pt). Scale bar in the SEM image: 500 μm

To address this, we changed a sentence in the revised manuscript:

Page 6, line 117: “Finally, all 400 devices from a 20×20 cross-bar array (see Supplementary Fig. 3) are measured to show spatial uniformity in terms of time constant and conductance change as a function of the number of applied pulses.”

Page 7, line 128: “The conductance map as a function of the device’s location in the optical image of the array shows uniform distribution conductance change, as shown in Fig. 1h.”

And we added the Figure R31 as a Supplementary Fig. 3 to show the clear image of the fabricated cross-bar array.

Comment #2:

2) the choice of anodizing technique and obtained TiO_x layer should be explained in more detail. What is the advantage in comparison with PVD or ALD? What phase was obtained (rutile, anatase or others? Is this nanocrystals or nanotubes. This answers should be addressed, I recommend provide X-ray analysis for phase identification.

Response: Thank you for these constructive comments. As the reviewer suggested, the reason for selecting the anodizing technique and the detailed description for the anodized TiO_x layer need to be described in the manuscript. The detailed responses for each comment are represented below.

(1) The choice of the anodizing technique over the PVD or ALD.

The anodizing technique is used to form TiO_x layer having gradient oxygen concentration. Anodizing is a top-down electrochemical oxidation method, and therefore, the Ti layer is oxidized vertically from the surface to the bottom. The reaction automatically finishes when the remaining Ti layer is oxidized and becomes TiO_x , because electrons for the reaction cannot be supplied through the layer. Until the reaction stops, the top region of the Ti layer goes through longer oxidation, while the bottom region of the layer experiences very short oxidation with limited electron and oxygen supply. Therefore, by using the anodizing technique, TiO_x layer with gradient oxygen concentration is easily formed at room temperature.

PVD such as sputtering or e-beam evaporation, or ALD can deposit TiO_x layer, but it is hard to make gradient oxygen concentration at room temperature through the PVD or ALD. For example, to deposit TiO_x layer with controlled oxygen concentration, it is necessary to use reactive sputtering, which usually requires a high temperature of about 400°C . Moreover, the TiO_x at room temperature with PVD does not show the self-rectifying property, because the Pt- TiO_x interface has lower barrier height compared to the anodized TiO_x case where the surface has nearly complete oxidized TiO_2 form. As shown in the Figure R32, a device with RF-sputtered TiO_2 at room temperature shows much larger leakage current than the anodized device. In the case of ALD, it can form a TiO_x film with better film quality and the oxygen concentration can be sophisticatedly controlled. In this work, we tried to introduce the advantage of the memristor device with oxygen gradient, and the anodizing technique is utilized because it is easy to form an oxide layer with oxygen gradient. Therefore, if the gradual TiO_x layer deposition is optimized in the ALD, it is expected that it would be another benefit of this work in terms of large-scale fabrication. The related work will be performed as the future work.

To clarify the reason of choosing the anodizing technique over the PVD, we added the Figure R32 and the corresponding sentences in Supplementary information as Supplementary Figure 5.

In addition, to inform that the other techniques including the ALD or the reactive sputtering can be used after optimizations for large scale memristor array fabrications, we added a sentence:

Page 23, line 518: “The anodizing process is used because it easily forms metal oxides with oxygen gradient, but other deposition methods including atomic layer deposition (ALD) or reactive sputtering could be used for fabrication of wafer-scale gradual TiO_x memristor devices.”

Figure R32. Consecutive 125 I-V curves of the Pt/sputtered TiO_x/Ti device. Instead of the anodized TiO_x layer, RF-sputtered TiO_x layer by sputtering a TiO_2 target at room temperature is utilized as insulating material to study the effect of the anodizing process. The thickness of sputtered TiO_x layer and anodized layer are both about 30 nm. Unlike the anodized device, the sputtered device shows unstable resistive switching with high leakage current and large variation. The leakage current increases in the sputtered device and the self-rectifying property deteriorates. In addition to the leakage, the large variation and the poor uniformity are induced from the evenly distributed oxygen in the sputtered TiO_x . It is hard to modulate the effective insulator thickness, because more oxygen should be moved to reduce the effective insulator in sputtered TiO_x case. Therefore, the sputtered device shows a small on/off ratio at the first cycle. After applying 125 voltage sweeps, the sputtered device shows a high on/off ratio similar to the anodized device. The consecutive voltage stimulus might change the oxygen distribution, which has a high oxygen concentration at the top electrode side and a low oxygen concentration at the bottom electrode side, similar to the anodized device.

(2) The phase and the structure of the anodized TiO_x layer

The developed TiO_x layer in this device is an amorphous thin film without nanotube or nanocrystalline structure. As the reviewer commented, it might be confusing whether the TiO_x layer is a nanotube or not, because the anodizing technique is widely used to form TiO_2 nanotubes. However, in this study, the electrolyte for anodizing does not make a nanotube structure, and thus the TiO_x film has a thin and uniform layer.

As the reviewer suggested, we conducted X-ray diffraction (XRD) test to the surface of the anodized TiO_x layer. The result about the phase of the TiO_x layer is shown in the Figure R33. From the test, we have revealed that the formed TiO_x layer has a fully amorphous structure, because there are no detected picks for rutile or anatase TiO_2 , which means that the anodized

TiO_x layer near the surface has a fully amorphous structure. This result corresponds to the other study that claims the anodizing makes a fully amorphous TiO₂ layer.

Figure R33. The results of the X-ray diffraction analysis (XRD) for the anodized TiO_x layer. The X-ray is illuminated on the surface of the layer; thus, the information about the surface structure is revealed. The results show that the anodized TiO_x has a fully amorphous structure near the surface region, by showing that no peak related to the crystal TiO₂ (e.g. rutile or anatase) is observed in the measured data.

To clearly represent the information about the structure of the anodized TiO_x layer, we added the Figure R33 as a Supplementary Figure 9, to show that the anodized TiO_x layer has a fully amorphous structure near the surface.

The corresponding sentences are added in the manuscript:

Page 7, line 144: “As shown in the Fig. 2a, the anodized TiO_x layer does not have any porous structure, even though the anodizing technique is widely used to form a nanoporous TiO₂ by anodizing Ti film in a fluoride-rich electrolyte.”

Page 9, line 186: “Moreover, in the anodized TiO_x layer, it is observed that the upper region of the TiO_x has a fully amorphous structure while the bottom region has a polycrystalline structure, because polycrystalline metallic Ti concentration is much higher in the bottom region of the TiO_x layer (see Supplementary Fig. 8 and 9).”

Comment #3:

3) authors should be more accurate in discussing mechanism of TiO₂ and conclusions (p. 9, word file). This paragraph sounds as a guess

Response: Thank you for your helpful comment. As the reviewer pointed out, the discussion about the device mechanism can be read as a guess. We have revised the manuscript to improve clarity.

First, a sentence “The size dependency in our results infers that the gradual TiO_x memristor operates by modulating the oxygen anion distribution in the overall TiO_x bulk.” Is modified, because the word “infers” sounds as a guess. The sentence is changed as:

Page 9, line 175: “The size dependency in the results verifies that the gradual TiO_x memristor operates by modulating the oxygen anion distribution in the overall TiO_x bulk.”

In addition, several words are replaced with more appropriate words. Therefore, we have changed or added following sentences:

Page 9, line 176: “In Supplementary Fig. 7, the device current reduces as the device size decreases due to the non-filamentary nature of the device.”

Page 9, line 180: “Furthermore, by calculating activation energy (E_A) from the decaying time constants under several elevated temperatures from 323 K to 423 K, we have proven that the oxygen anion movement across the whole TiO_x bulk dominates resistive switching (see Fig. 2f).”

Comment #4:

4)20 is not very high on/of ratio. What is on/of ratio for individual memristor? Is 20 is average for all arrays?

Response: Thank you for those valuable comments. The value is obtained after applying 4 pulses. The maximum on/off ratio of the device from the linear voltage sweep I-V curve is more than 2,700 as shown in Figure R34.

Figure R34. A device I-V hysteresis curve (left), and an extracted on/off ratio of the device for each voltage point from the I-V curve (right). As shown in the right figure, the on/off ratio has the highest value ($>2,700$) when about 1 V is applied to the device.

To clarify this information, we modify the original sentence as following:

Page 16, line 341: “This experiment proves that the gradual TiO_x memristor as a reservoir can separate different binary sequences with high uniformity in an array platform. It is noticeable that the devices show high on/off ratio (~ 20) with only four pulses in the Fig. 4a. The maximum on/off ratio from the developed memristor’s large on/off ratio ($>2,700\times$), as shown in Supplementary Fig. 15.”

We added the Figure R34 in Supplementary information as a Supplementary figure 15.

Comment #5:

Fig. S4 provides only 2 cycles, for better demonstration of stability more cycles (at least 50, if authors claims 50 cycles at fig. 1 and 10^6 cycles at p.6).

Response: Thank you for the constructive suggestion. To address this comment, we have measured the sputtered device again in the same manner as the Figure 1b in the manuscript. The I-V curve from the sputtered device with consecutive 125 linear voltage sweeps is shown in figure R35. On the contrary to the anodized device, the sputtered device shows high variation during the cycles. In the sputtered device, there is uniformly distributed oxygen concentration across the TiO_2 layer. By applying the e-field to the device, the oxygen distribution in the oxide layer changes. Therefore, the device characteristics are changed by the consecutive voltage sweeps. In addition, the sputtered device shows a small on/off ratio at the first cycle. After applying 125 voltage sweeps, the sputtered device shows a high on/off ratio similar to the anodized device. The consecutive voltage stimulus might change the oxygen distribution, which has a high oxygen concentration at the top electrode side and a low oxygen concentration at the bottom electrode side, similar to the anodized device.

Figure R35. Consecutive 125 I-V curves of the Pt/sputtered TiO₂/Ti device. Instead of the anodized TiO_x layer, RF-sputtered TiO_x layer by sputtering a TiO₂ target at room temperature is utilized as an insulating material to study the effect of the anodizing process. The thickness of sputtered TiO_x layer and anodized layer are both about 30 nm. Unlike the anodized device, the sputtered device shows unstable resistive switching with high leakage current and large variation. The leakage current increases in the sputtered device and the self-rectifying property deteriorates. In addition to the leakage, the large variation and the poor uniformity are induced from the evenly distributed oxygen in the sputtered TiO_x. It is hard to modulate the effective insulator thickness, because more oxygen should be moved to reduce the effective insulator for the sputtered TiO_x case. Therefore, the sputtered device shows a small on/off ratio at the first cycle. After applying 125 voltage sweeps, the sputtered device shows a high on/off ratio similar to the anodized device. The consecutive voltage stimulus might change the oxygen distribution, which has a high oxygen concentration at the top electrode side and a low oxygen concentration at the bottom electrode side, similar to the anodized device.

To clarify this point, we have replaced the original figure with the Figure R35 as Supplementary Fig. 5. The original figure contains only two cycles but the revised figure contains consecutive 125 cycles.

The corresponding explanation about the Figure R35 is also modified to emphasize the non-uniformity of the sputtered device:

Page 8, line 154: “This oxygen gradient effectively controls the thickness of the insulating region and results in resistive switching with a large on/off ratio of the anodizing TiO_x layer, while the sputtered TiO₂ layer cannot effectively change the thickness of the insulating region due to uniform oxygen concentration and shows a low on/off current ratio and a non-uniform switching characteristic (see Supplementary Fig. 5).”

Comment #6:

5) p. 13, clarification for mechanism of killing cell should be provided "The positively charged hydrophilic portion of the peptide attaches to the negatively charged surface of the bacteria, and the hydrophobic portion interacts with proteins on the surface of the bacterial cell to induce “bacterial lysis”, which kills the cell"

Response: We thank the reviewer for this fruitful suggestion. The original explanation for the mechanism of AMP is complicated, and thus, we have changed the sentence to depict the mechanism clearly.

To address this point, we have changed the sentence about the bacteria-killing mechanism in the revised manuscript:

Page 19, line 398: “The AMPs are aggregated and inserted into the bacterial epidermis (surface of the bacteria). After that, because the bacterial epidermis is hydrophobic while the environment is hydrophilic, the hydrophobic side of the AMP is attached to the bacterial epidermis. The hydrophilic side of AMPs are faced with each other, and then the aggregated

AMPs finally form a pore structure across the bacterial epidermis. Because of the pore consisting of AMPs, the cytoplasm of the bacteria flows outside and the bacteria is killed. This cell death process induced by AMPs is called bacterial lysis.”

Reference:

1. Lim, H. *et al.* Reliability of neuronal information conveyed by unreliable neuristor-based leaky integrate-and-fire neurons: A model study. *Sci. Rep.* **5**, 1–15 (2015).
2. Wang, Z. *et al.* Fully memristive neural networks for pattern classification with unsupervised learning. *Nat. Electron.* **1**, 137–145 (2018).
3. Duan, Q. *et al.* Spiking neurons with spatiotemporal dynamics and gain modulation for monolithically integrated memristive neural networks. *Nat. Commun.* **11**, 1–13 (2020).
4. Morris, R. G. M. D.O. Hebb: The Organization of Behavior, Wiley: New York; 1949. *Brain Res. Bull.* **50**, 437 (1999).
5. Wang, Z. *et al.* Fully memristive neural networks for pattern classification with unsupervised learning. *Nat. Electron.* **1**, 137–145 (2018).
6. Wang, Z. *et al.* Memristors with diffusive dynamics as synaptic emulators for neuromorphic computing. *Nat. Mater.* **16**, 101–108 (2017).
7. Zhang, X. *et al.* An Artificial Neuron Based on a Threshold Switching Memristor. *IEEE Electron Device Lett.* **39**, 308–311 (2018).
8. Kumar, S., Williams, R. S. & Wang, Z. Third-order nanocircuit elements for neuromorphic engineering. *Nature* **585**, 518–523 (2020).
9. Duan, Q. *et al.* Spiking neurons with spatiotemporal dynamics and gain modulation for monolithically integrated memristive neural networks. *Nat. Commun.* **11**, 1–13 (2020).
10. Yi, W. *et al.* Biological plausibility and stochasticity in scalable VO₂ active memristor neurons. *Nat. Commun.* **9**, (2018).
11. Jerry, M., Parihar, A., Grisafe, B., Raychowdhury, A. & Datta, S. Ultra-low power probabilistic IMT neurons for stochastic sampling machines. *IEEE Symp. VLSI Circuits, Dig. Tech. Pap.* **8**, T186–T187 (2017).
12. Du, C. *et al.* Reservoir computing using dynamic memristors for temporal information processing. *Nat. Commun.* **8**, 1–10 (2017).
13. Moon, J. *et al.* Temporal data classification and forecasting using a memristor-based reservoir computing system. *Nat. Electron.* **2**, 480–487 (2019).
14. Midya, R. *et al.* Reservoir Computing Using Diffusive Memristors. *Adv. Intell. Syst.* **1**, 1900084 (2019).

15. Zhong, Y. *et al.* Dynamic memristor-based reservoir computing for high-efficiency temporal signal processing. *Nat. Commun.* **12**, 1–9 (2021).
16. Sun, L. *et al.* In-sensor reservoir computing for language learning via two-dimensional memristors. *Sci. Adv.* **7**, (2021).
17. El Sallab, A., Abdou, M., Perot, E. & Yogamani, S. Deep reinforcement learning framework for autonomous driving. *IS T Int. Symp. Electron. Imaging Sci. Technol.* 70–76 (2017) doi:10.2352/ISSN.2470-1173.2017.19.AVM-023.
18. Zhang, L., Tan, J., Han, D. & Zhu, H. From machine learning to deep learning: progress in machine intelligence for rational drug discovery. *Drug Discov. Today* **22**, 1680–1685 (2017).
19. Dean, J. The Deep Learning Revolution and Its Implications for Computer Architecture and Chip Design. *Dig. Tech. Pap. - IEEE Int. Solid-State Circuits Conf.* **2020-Febru**, 8–14 (2020).
20. Jumper, J. *et al.* Highly accurate protein structure prediction with AlphaFold. *Nature* **596**, 583–589 (2021).
21. Indiveri, G. *et al.* Neuromorphic silicon neuron circuits. *Front. Neurosci.* **5**, 1–23 (2011).
22. Choi, S. *et al.* SiGe epitaxial memory for neuromorphic computing with reproducible high performance based on engineered dislocations. *Nat. Mater.* **17**, 335–340 (2018).
23. Lee, M. J. *et al.* A fast, high-endurance and scalable non-volatile memory device made from asymmetric Ta₂O_{5-x}/TaO_{2-x} bilayer structures. *Nat. Mater.* **10**, 625–630 (2011).
24. Yeon, H. *et al.* Alloying conducting channels for reliable neuromorphic computing. *Nat. Nanotechnol.* **15**, 574–579 (2020).

REVIEWER COMMENTS

Reviewer #1 (Remarks to the Author):

In this research, the authors fabricate a volatile RRAM device which shows low device-to-device and cycle-to-cycle variability and high dynamic range. The device does not require forming and can be arranged in a crossbar without any selector device, which makes the device energy- and area-efficient. The authors show that the volatile nature of the device can be leveraged to design leaky integrate and fire neurons. The authors also demonstrate that the decay in the device can be modulated with pulses with different duty cycles. This property is leveraged to design a memristor reservoir computing system.

The authors have extensively edited the manuscript. The new analysis into time-constant and W/I ratio was helpful and improved the contributions. The additional information in relation to a biological neuron is in the right direction. Last, several comparisons and points provided in the rebuttal helped clarify the aim of this paper and how it can be beneficial.

After reviewing the authors extensive rebuttal on the contribution towards providing a comprehensive memristive RC system that overcomes several key limitations in both the synaptic and neuron components, there are several key unanswered questions. The authors incorporated new content on how this device can be used to implement artificial neurons in neuromorphic computing. The previous responses have been suggestions to find a novel application, such as acting as the integration of membrane potential rather than a capacitor. It is not clear what advantages in comparison to other memristive-based neurons the presented device offers.

The tables comparing devices presented in the rebuttal are helpful, however the neuron-based comparison, does not truly offer any insight into where/why a particular device may be better for neuron implementations. For the RC-based comparison, typically RC systems use memristors as a synapse which the volatile memristor cannot support. Also, the discussion on biological neurons in relation to the proposed device is not clear. Typically the term short-term plasticity refers to modeling dynamic changes of the synaptic efficacy caused by a neurons recent firing (when a neuron emits an action potential it will release neurotransmitters, these neurotransmitters take time to replenish and further action potentials in a short time-window will have reduced impact). It is not clear what is being modeled, or how it is beneficial to the proposed RC system and tested applications. The results presented show the device is only used as a linear neuron with a temporal component and does not involve any other use of memristors, which disconnects the neuron discussion from the rest of the paper. For the synapse, a weight still requires non-volatile memristors rather than the device proposed here which would suffer from the aforementioned bottlenecks (yield, sneak paths, etc.).

Algorithmically, the model is not representative of a typical RC network. The proposed implementation translates an input sequence into a one-hot encoded vector (standard practice), but then projects this vector in a 1-to-1 mapping onto memristors in the reservoir column. Then several rows of devices are instantiated with different properties to capture different temporal dynamics of the input. Though an

interesting approach, it is not truly RC as there is no random projection of an input space into a high-dimensional output space. Rather it is a temporally filtered input, using different time-constants, passed directly to a readout layer. This approach would likely not work for many applications, and is tested in a very narrow scope with little to compare to. The references pointed out in the rebuttal to similar approaches, do not seem to constitute a good baseline for the RC networks. On the other hand, if the focus is on the best AMP generation using an efficient neuromemristive system (not an RC) then this approach would be more valuable as it would show there is no need for additional complexity to solve the problem, and the proposed device can achieve this in an efficient manner.

Reviewer #2 (Remarks to the Author):

The authors have largely addressed my concerns. The revisions, along with the new data on the neuron circuits, made the paper stronger. I thus support its publication in Nature Communications.

Reviewer #3 (Remarks to the Author):

The authors answered the questions addressed: both experimentally and in text. I recommend publishing the article as it is

Response Letter to Reviewers' Comments

We sincerely appreciate the reviewers for investing their valuable time and effort on reviewing our manuscript and providing insightful comments and suggestions to help further improve the quality of our work. Considering the reviewers' evaluations, we have made a point-by-point response to the reviewer 1's comments, and revised our manuscript to improve the clarity of our work. We believe we have addressed the reviewer 1's comments and now the paper is more rigorous in content and clearer in presentation. Based on some responses below, we have appended the modelling of biological neuron system using our artificial memristive neuron and modified the tables for neuron comparison. Our point-by-point responses to the reviewer 1's comments are as follows.

Reviewer #1

In this research, the authors fabricate a volatile RRAM device which shows low device-to-device and cycle-to-cycle variability and high dynamic range. The device does not require forming and can be arranged in a crossbar without any selector device, which makes the device energy- and area-efficient. The authors show that the volatile nature of the device can be leveraged to design leaky integrate and fire neurons. The authors also demonstrate that the decay in the device can be modulated with pulses with different duty cycles. This property is leveraged to design a memristor reservoir computing system.

The authors have extensively edited the manuscript. The new analysis into time-constant and W/I ratio was helpful and improved the contributions. The additional information in relation to a biological neuron is in the right direction. Last, several comparisons and points provided in the rebuttal helped clarify the aim of this paper and how it can be beneficial.

Response: We sincerely thank the reviewer for positive comments on our revised manuscript. Based on the reviewer's comments and suggestions, we have improved several sentences and added a table emphasizing the advantages of the proposed memristor device for artificial neuron compared to other existing neuron devices. For the reservoir computing section, we have changed the terminology from "reservoir computing" to "neuro-memristive computing" for sequence data processing as the reviewer suggested. Thanks for the reviewer's valuable comments, now the advantages of the device are described clearly. Our detailed responses to the reviewer's comments are provided below.

Comment #1:

After reviewing the authors extensive rebuttal on the contribution towards providing a comprehensive memristive RC system that overcomes several key limitations in both the synaptic and neuron components, there are several key unanswered questions. The authors incorporated new content on how this device can be used to implement artificial neurons in neuromorphic computing. The previous responses have been suggestions to find a novel application, such as acting as the integration of membrane potential rather

than a capacitor. It is not clear what advantages in comparison to other memristive-based neurons the presented device offers.

Response: We would like to thank the reviewer for this valuable comment. The proposed memristor device has high device-to-device and cycle-to-cycle uniformity, which have been the main bottlenecks of memristor devices being used today. Because memristive neuron device processes inputs and transfers the processed data to the next layer, the uniformity of the neuron device is essential. However, other existing neuron devices such as diffusive memristors and Mott memristors have shown large variation, and the device-to-device and cycle-to-cycle uniformity of these devices have not been quantitatively analyzed. In addition, the existing neuron devices have been tested on the stand-alone device only, and therefore, it is hard to assess the feasibility of the neuron device to be operational as a large scale crossbar array. In our devices, thanks to the low variation ($< 1.39\%$ c-to-c and $< 3.87\%$ d-to-d) of the proposed memristor device, we have built a 20×20 memristor cross-bar array without the sneak path issue. This is the main advantage of our device compared to other memristor-based neurons.

In addition, as the reviewer suggested, the proposed device can be used for integrating the membrane potential instead of using a capacitor. The memristive artificial neurons usually consist of a memristor and a capacitor in parallel. The capacitor integrates the membrane potential, and the artificial neuron generates spikes when the potential reaches the memristor's set voltage. However, a capacitor requires large area, thus it degrades the scalability of the artificial neuron. The proposed gradual TiO_x memristor can perform the integration of membrane potential instead of a capacitor. If the gradual TiO_x memristor is connected to the volatile memristor (e.g. diffusive memristor or Mott memristor), the applied voltage to volatile memristor can be decided by the resistance ratio between the two devices. As the resistance of the gradual TiO_x memristor gradually decreases, the voltage across the volatile memristor increases. Once the resistance of the gradual TiO_x memristor becomes low enough, the high voltage across the volatile memristor induces abrupt switching followed by spike generation.

We have verified this approach experimentally and the results are shown in the Figure R1. The gradual TiO_x memristor is serially connected to the diffusive memristor ($\text{Ag}/\text{Al}_2\text{O}_3/\text{p}^+\text{-Si}$). It shows leaky integrate-and-fire (LIF) behavior as shown in Figure R1b and R1c, similar to a typical artificial neuron with a capacitor. This approach enables a small size (4F^2) artificial neuron to operate without a large size capacitor.

This approach still needs further improvement due to the large variation/yield problem of conventional volatile memristors, including diffusive memristors and the Mott memristors. Therefore, we are preparing this study (artificial neuron without a capacitor) as future work.

Figure R1. Comparison between the conventional memristor based artificial neuron and newly developed artificial neuron without capacitor, and the experimental result of the new approach. a, An illustration of the structure of conventional memristor based artificial neuron, which consist of a volatile memristor and a capacitor in parallel. **b,** An illustration of the structure of new memristor based artificial neuron which consists of a gradual TiO_x memristor and a volatile memristor in serial. The gradual TiO_x memristor is used as an membrane potential integrator instead of a capacitor **c,** The experimental result of the newly developed memristor based artificial neuron in (b). The neuron shows leaky-integrate and fire operation similar to the conventional memristor based artificial neurons.

Comment #2:

The tables comparing devices presented in the rebuttal are helpful, however the neuron-based comparison, does not truly offer any insight into where/why a particular device may be better for neuron implementations. For the RC-based comparison, typically RC systems use memristors as a synapse which the volatile memristor cannot support. Also, the discussion on biological neurons in relation to the proposed device is not clear. Typically the term short-term plasticity refers to modeling dynamic changes of the synaptic efficacy caused by a neurons recent firing (when a neuron emits an action potential it will release neurotransmitters, these neurotransmitters take time to replenish and further action potentials in a short time-window will have reduced impact). It is not clear what is being modeled, or how it is beneficial to the proposed RC system and tested applications. The results presented show the device is only used as a linear neuron with a temporal component and does not involve any other use of memristors, which disconnects the neuron discussion from the rest of the paper. For the synapse, a weight still requires

non-volatile memristors rather than the device proposed here which would suffer from the aforementioned bottlenecks (yield, sneak paths, etc.).

Response: We appreciate the reviewer for the constructive comments. Detailed explanations are shown below.

i) About neuron comparison

As the reviewer commented, the table for comparing neuron characteristics does not clearly show the advantages of the proposed memristor. Reliable operation of a neuron is a necessity because a malfunction of a single neuron device can cause the degradation of the entire system performance. Even though several volatile memristor based neurons have been studied, they have suffered from large variations, and quantitative information about the uniformity is not given. The proposed neuron device in this work shows a high cycle-to-cycle (1.39%) and device-to-device uniformity(3.87%) from a 20×20 crossbar array. Therefore, the proposed device can be used for the large-scale implementation of memristor based neuromorphic computing system.

To emphasize the advantages of the proposed device on the uniformity and reliability for neuromorphic computing, we have modified supplementary table 1 as below (please see the table R1). We have added the fabricated structure of the neuron devices to clearly show the advantage of the proposed device for large-scale integration, and we also have removed some columns (endurance, spike current, and off-resistance) in the table that do not show clear advantages. Table R1 is appended in the revised supplementary manuscript.

Device structure	Operation mechanism	Cycle-to-cycle variation	Device-to-device variation	Fabricated structure
Pt/gradual TiO _x /Ti	Ionic (oxygen vacancies)	1.39%	3.87%	20×20 crossbar array
Pt/SiO _x N _y :Ag/Pt ^{1,2}	Ionic (diffusion of Ag)	N/A	N/A	1×8 array
Ag/SiO ₂ /Au ³	Ionic (diffusion of Ag)	N/A	N/A	Stand-alone device
Pt/TiN/NbO ₂ /TiN/W ⁴	Mott	Good, but not quantitatively analyzed	N/A	1×24 array
Pt/Ti/NbO ₂ /Pt/Ti ⁵	Mott	Good, but not quantitatively analyzed	N/A	1×4 array
Pt/VO ₂ /Pt ⁶	Mott	Good, but not quantitatively analyzed	7%	Stand-alone device
TE/VO ₂ /BE ⁷	Mott	N/A	N/A	Stand-alone device

Table R1. Comparisons with various memristor-based artificial neurons. The cycle-to-cycle and device-to-device uniformities of the gradual TiO_x memristor are compared to the other existing neuron memristors (diffusive memristors and Mott memristors). The gradual TiO_x memristor shows superior uniformities in both cycle-to-cycle and device-to-device compared to the other neuron memristors. In addition, the gradual TiO_x memristor can be integrated in the cross-bar array form thanks to its superior reliability, while the others hardly show cross-bar array integration.

ii) About RC comparison

We thank the reviewer for the correction. For comparison, the proposed device have been used as a sequential data processor, not as a synapse. The current platform cannot be used for random projection of an input space into a high-dimensional output space, and it is not a typical reservoir computing. As the reviewer claimed, we have removed “reservoir computing” terminology and used “neuro-memristive computing” instead. We also removed the general explanation about RC from the manuscript. Detailed explanation about the RC part will be discussed in the reviewer’s comment 3.

iii) Discussion on short-term plasticity terminology.

As the reviewer commented, the usage of the term ‘short-term plasticity’ in the manuscript is not suitable. Therefore, we have removed Figure 3b (about short-term plasticity from synaptic behavior) in the original manuscript to prevent any misunderstanding. We have also modified the term ‘short-term plasticity’ into ‘short-term memory effect’ to clarify the expressions in the neuro-memristive computing work.

Comment #3:

Algorithmically, the model is not representative of a typical RC network. The proposed implementation translates an input sequence into a one-hot encoded vector (standard practice), but then projects this vector in a 1-to-1 mapping onto memristors in the reservoir column. Then several rows of devices are instantiated with different properties to capture different temporal dynamics of the input. Though an interesting approach, it is not truly RC as there is no random projection of an input space into a high-dimensional output space. Rather it is a temporally filtered input, using different time-constants, passed directly to a readout layer. This approach would likely not work for many applications, and is tested in a very narrow scope with little to compare to. The references pointed out in the rebuttal to similar approaches, do not seem to constitute a good baseline for the RC networks. On the other hand, if the focus is on the best AMP generation using an efficient neuromemristive system (not an RC) then this approach would be more valuable as it would show there is no need for additional complexity to solve the problem, and the proposed device can achieve this in an efficient manner.

Response: Thanks for this insightful comment. We agree with the reviewer’s suggestion that the used model is not a representative of a RC network, even though it can be useful for temporal/sequential data processing. Therefore, we now focus on AMP generation application using an efficient neuro-memristive computing that processes sequential information based on the short-term memory effect of the proposed device. We have removed the descriptions about the reservoir computing from the original manuscript, and changed the terminology from “reservoir computing” to “neuro-memristive computing”.

Reviewer #2

The authors have largely addressed my concerns. The revisions, along with the new data on the neuron circuits, made the paper stronger. I thus support its publication in Nature Communications.

Response: We highly appreciate the reviewer for the valuable time for reviewing our manuscript and constructive comments to help significantly improve the quality of our work.

Reviewer #3

The authors answered the questions addressed: both experimentally and in text. I recommend publishing the article as it is

Response: We would like to sincerely thank the reviewer for the valuable time the reviewer has spent reviewing our manuscript and providing insightful comments which greatly improve the quality of our work.